# Designing Observation and Action Models for Efficient Reinforcement Learning with LLMs

## Abstract

The design of observation and action models is a fundamental step in reinforcement learning (RL), as it defines how agents perceive and interact with their environment. Despite its importance, this design choice is often overlooked in standard benchmarks, which typically use handcrafted models. However, this choice can substantially influence both learning efficiency and final performance. To address this gap, we propose LOAM (LLM-based design of Observation and Action Models), a framework that leverages large language models (LLMs) to automate the generation of these models. LOAM extracts information about simulation variables and task objectives from the environment and uses an LLM to generate Python functions for the observation and action models, enabling seamless integration into standard RL training pipelines. When applied to the basic locomotion tasks of HumanoidBench (stand, walk, run), LOAM-designed models achieve over 3× faster learning on average with the same FastTD3 algorithm compared to the default benchmark models. Furthermore, to handle the variability of LLM outputs, we race multiple generated designs and progressively select the top performers under a fixed training budget. To our knowledge, this is the first work to propose an efficient learning method that mitigates quality diversity in LLM-designed observation and action models within the same timestep as the standard single-model training.

## 1 Introduction

Reinforcement learning (RL) has achieved remarkable success across diverse domains, from mastering complex board games (Silver et al., 2016; 2017; 2018) to controlling simulated and real-world systems (Tassa et al., 2018; James et al., 2020; Mittal et al., 2023; Pérez-D'Arpino et al., 2021; Chuck et al., 2024). While recent advances have enabled RL agents to successfully handle various control tasks, a significant challenge remains in environments with complex observation and action spaces, which are increasingly common in practical applications where agents must process diverse information sources for effective task completion.

Practitioners have traditionally relied on handcrafted observation and action models to manage this complexity, where domain experts manually select relevant features from raw sensory data and design appropriate control abstractions. However, this manual design process represents a fundamental bottleneck: the resulting models are fixed in advance and may substantially influence both learning efficiency and final performance. Alternative observation representations might capture task-relevant information more efficiently, while different action parameterizations could provide control interfaces that are more amenable to policy optimization. Despite their potential to accelerate training and improve final performance, exploring alternative designs remains prohibitively costly due to the substantial engineering effort from domain experts.

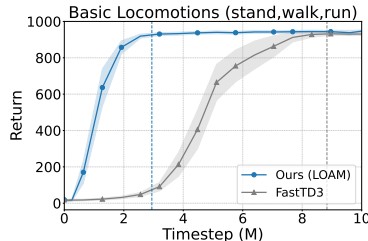

Figure 1: LOAM is 3× faster than default benchmark model (FastTD3) across HumanoidBench stand, walk, and run tasks.

To address this challenge, we introduce **LOAM** (**L**LM-based design of **O**bservation and **A**ction **M**odels), an automated framework for generating observation and action representations in complex RL environments. By employing large language models (LLMs), LOAM analyzes environment specifications and task requirements to produce Python implementations of observation and action models that directly integrate with existing RL training workflows. Additionally, we present LOAM-Race, which addresses the variability of LLM outputs by evaluating multiple candidate models in parallel, progressively selecting the top performers within a fixed training budget to identify optimal observation and action models.

Figure 1 demonstrates the averaged performance on the basic locomotion tasks (stand, walk, run) in the challenging Humanoid benchmark (Sferrazza et al., 2024). We compare two configurations: (1) the state-of-the-art FastTD3 algorithm (Seo et al., 2025) using handcrafted features provided by the default benchmark (gray), and (2) LOAM combined with FastTD3 with access to all available raw features (blue). Remarkably, LOAM-designed models achieve over 3× faster learning on average with the same FastTD3 algorithm compared to the default benchmark models.

This work makes three primary contributions:

1. **LLM-Designed Observation and Action Models Based on Task Description:** We propose **LOAM**, a framework that automatically generates task-relevant observation and action models for RL agents. LOAM leverages LLMs to generate Python functions for these models, enabling seamless integration into standard RL training pipelines.

2. **Robust Model Selection with LOAM-Race:** We introduce **LOAM-Race**, a racing mechanism that efficiently evaluates multiple LOAM-generated models in parallel and adaptively allocates resources to identify high-performing models. To our knowledge, this is the first work to propose an efficient learning method that mitigates quality diversity in LLM-designed observation and action models within the same timestep as the standard single-model training.

3. **State-of-the-Art Performance on HumanoidBench:** We demonstrate that LOAM-designed models achieve state-of-the-art performance on humanoid control tasks in HumanoidBench— a challenging benchmark with numerous sensors and actuators— substantially surpassing expert-designed baselines in both learning speed and final performance.

## 2 BACKGROUND

### 2.1 RL PROBLEM FORMULATION

Reinforcement learning (RL) addresses sequential decision-making problems where an agent interacts with an environment to maximize long-term rewards. In practice, agents do not have access to the full underlying state but instead rely on *raw observation features* that are often incomplete, noisy, and redundant (Nguyen et al., 2023). This motivates the Partially Observable Markov Decision Process (POMDP) formalism (Sutton et al., 1998; Kaelbling et al., 1998), defined as:

$$\mathcal{M} = \langle S, O, A, P, \Omega, R, \rho, \gamma \rangle,$$

where $S$ is the state space, $O$ the observation space, $A$ the action space, $P(s'|s,a)$ the transition dynamics, $\Omega(s)$ the observation function, $R(s,a)$ the reward function, $\rho(s_0)$ the initial state distribution, and $\gamma$ the discount factor. The goal is to find a policy $\pi(a|o)$ that maximizes the expected discounted return:

$$\pi^* = \arg\max_{\pi} \mathbb{E}\Big[ \sum_{t=0}^{\infty} \gamma^t R(s_t, a_t)\Big].$$

Simulation benchmarks such as MuJoCo (Todorov et al., 2012), MuJoCo Playground (Zakka et al., 2025), and HumanoidBench (Sferrazza et al., 2024) instantiate this formalism in high-dimensional environments. MuJoCo has long served as a standard platform for continuous control, while more recent benchmarks such as MuJoCo Playground and HumanoidBench advance evaluation toward complex humanoid control, exposing the challenges of learning with hundreds of sensors and actuators. A common limitation, however, is that these environments typically provide fixed observation and action models, restricting the study of how different specifications influence learning.

Among them, HumanoidBench is particularly notable for offering diverse locomotion and manipulation tasks, making it a physically plausible yet challenging testbed that we adopt as our primary experimental environment.

## 2.2 DESIGNS OF OBSERVATION AND ACTION MODELS

Observation and action models specify how the agent perceives the environment and executes actions. Giving access to the entire set of raw observation features and full action commands—the *raw observation space* $O_{raw}$ and the *raw action space* $A_{raw}$—provides complete information but results in prohibitively high-dimensional spaces, leading to increased sample complexity.

**Observation models** reduce the raw observation space to a smaller set of task-relevant features. Formally, they define a mapping

$$f_o : O_{raw} \to O^{f_o},$$

which transforms raw observation features $o_{raw} \in O_{raw}$ into compact representations $f_o(o_{raw}) \in O^{f_o}$ for policy learning $\pi(a|f_o(o_{raw}))$. This mapping $f_o$ is typically handcrafted in common benchmarks; for instance in robot control problems, observations are fixed to include joint positions and velocities, contact indicators, or actuator states, while global positions are excluded to enforce translational invariance (Todorov et al., 2012; Zakka et al., 2025; Sferrazza et al., 2024).

**Action models** either use the raw action space directly or introduce a mapping to transform high-level actions into full action commands. Formally, they define a mapping

$$f_a : A^{f_a} \to A_{raw},$$

which expands high-level actions into full action commands. In many benchmarks the raw action space is used directly, while in some cases (Sferrazza et al., 2024; Makoviychuk et al., 2021) $f_a$ is manually specified with simple rules such as normalization, constraints, or joint coupling.

Thus, widely used environments rely on handcrafted $f_o$ and $f_a$ to make training feasible. While this standardization enables consistent evaluation, it also fixes the design of observation and action models in advance. As a result, the impact of alternative designs on learning efficiency and performance remains largely unexplored, motivating automated approaches that can generate such models across diverse tasks.

## 3 LOAM: LLM-BASED DESIGN OF OBSERVATION AND ACTION MODELS

Large language models (LLMs) can be used to automatically design observation and action models for RL environments. Instead of relying on handcrafted specifications, structured prompts guide the LLM to generate executable Python functions. These functions transform raw observations into compact, task-relevant representations and map designed actions back to the full action commands of the environment.

### 3.1 STRUCTURED PROMPTS FOR MODEL DESIGN

The prompt structure consists of three categories: *system prompts*, *observation model prompts*, and *action model prompts*. To generate observation and action models, the system prompts are combined with the observation and action model prompts, respectively. Table 1 presents the structural components of the prompts. For the complete prompts, please refer to Appendix F.4.

Our experiments indicate that incorporating guidance on whole-body posture stability substantially in the reasoning and planning prompts improves model quality. This result suggests that stability considerations support task success in robotics domains, as demonstrated in our experimental results in Appendix F.4.

### 3.2 LLM-GENERATED PYTHON FUNCTIONS FOR OBSERVATION AND ACTION MODELS

Given the structured prompts described above, the LLM produces executable Python code that implements either an observation model function $f_o$ or an action model function $f_a$. Observation models are generated as `compute_obs` functions that extract task-relevant features from the full

Table 1: Structure of LOAM's Model Prompts

| Prompt Type | Components | Description |
| --- | --- | --- |
| System Prompt | Role and Objective | LLM's persona as RL researcher and design objective |
| | Task Description | Name, objective, initialization & termination of the given task |
| | Available Variable Info | Raw observation/action attributes: names, shapes, roles, indices |
| Observation/Action Model Prompt | Instruction | Function signature with input/output specifications |
| | Reasoning & Planning | Step-by-step task analysis and feasibility verification guidelines |
| | Action Mapping (Action model only) | Action dimension specifications |
| | Output Guide | Python function format guidelines |

observation space, while action models are generated as compute_action functions that map the low-dimensional output from the policy network to the full set of action commands. These functions define the key transformations between raw and designed state–action spaces, thereby enabling more efficient policy learning in complex environments.

## 4 Reinforcement Learning with LOAM-Designed Models

The previous section introduced LOAM for generating observation and action models. We now show how these models are integrated into RL environments and used for efficient policy learning.

### 4.1 Integrating LOAM into RL Environments

LOAM produces two Python functions, compute_obs and compute_action, which implement the observation model $f_o$ and the action model $f_a$. These functions wrap the original environment to form a re-modeled RL environment with newly defined observation and action spaces.

**LOAM-Based Environment Wrapper**

```python
class ReModelEnv(RLEnv):
    def __init__(self, env, compute_obs, compute_action):
        self._env = env
        self._compute_obs = compute_obs
        self._compute_action = compute_action

    def step(self, modeled_act):
        full_act = self._compute_action(modeled_act)
        full_next_obs, reward, done, info = self._env.step(full_act)
        return self._compute_obs(full_next_obs), reward, done, info
```

The training loop proceeds as follows. At each timestep, the agent receives a modeled observation and outputs a modeled action that is transformed into a complete action command through compute_action for the original environment. After the environment executes this action, compute_obs transforms the resulting raw observation back into a modeled observation for the agent. This cycle repeats until reaching the maximum timesteps, enabling seamless policy learning with LOAM-designed models.

### 4.2 LOAM-Race: Racing Multiple Designs for Robust Model Selection

While observation and action models generated by LOAM typically exhibit rapid learning progress, policies trained on them can occasionally perform poorly due to the inherent stochasticity of LLMs.

---

**Algorithm 1** LOAM-Race

---

**Input:** $K$: number of environment models, $f$: acquisition function, $T$: maximum timesteps
**Output:** The best policy $\pi^*$ and environment model $\mathcal{M}^*$
 1: Generate $K$ environment models $\mathcal{M}_k, k = 1, \ldots K$ using LOAM
 2: Initialize return histories $\mathcal{R}_k, k = 1, \ldots K$
 3: $i \leftarrow 1$
 4: **while** total timesteps $< T$ **do**
        *# Allocate training resources for $i$-th model*
 5:     Rollout $\pi_i$ on $\mathcal{M}_i$ for race timesteps and update returns to $\mathcal{R}_i$
 6:     Train $\pi_i$ with collected experiences
        *# Select most promising model using acquisition score*
 7:     $i \leftarrow \arg\max_k f(\mathcal{R}_k)$
 8: **end while**
    *# Select the best performing policy and its environment model*
 9: $i \leftarrow \arg\max_k \max(\mathcal{R}_k)$
10: **return** $\pi_i, \mathcal{M}_i$

---

Prior works (Ma et al., 2023; Wang et al., 2024) address this issue by generating multiple candidate models and selecting the best-performing one. However, identifying the optimal candidate imposes a significant computational burden, as it typically necessitates training each candidate policy to convergence to accurately assess its performance. For instance, LESR (Wang et al., 2024) evaluates 18 candidate models by training each for $0.8T$, where $T$ represents the timestep budget for a single-model training. Following this, the selected best model undergoes a full training ($1T$). Consequently, this approach consumes $18 \times 0.8T + 1T = 15.4T$ in total—roughly 15 times the cost of training a policy on a single model—which substantially degrades sample efficiency.

We introduce LOAM-Race to address this limitation. The rationale behind LOAM-Race is that the future potential of a candidate model can be estimated without training its policy to convergence, enabling efficient model selection with significantly reduced computational cost. A naive selection criterion would simply choose the model with the highest current return. However, early-stage returns are noisy estimates of final performance, and a model that lags initially may eventually surpass others.

To account for this uncertainty, we adopt the principle of *Optimism in the Face of Uncertainty (OFU)* (Jamieson et al., 2014; Auer et al., 2002; Maron & Moore, 1993): instead of selecting based on current performance alone, we select the model with the highest upper confidence bound on predicted final performance. This acquisition function balances exploitation (favoring currently strong models) with exploration (giving uncertain models a chance to prove themselves). To compute these upper confidence bounds—serves as the acquisition score for each candidate, we extrapolate each candidate's return trajectory using *Bayesian ridge regression*.

The LOAM-Race algorithm is outlined in Algorithm 1. To select the most promising model, LOAM-Race computes an acquisition score for each candidate. We define the race timestep as the minimum number of timesteps required to obtain sufficient information for computing these scores. In practice, we set the race timestep to approximately $T/20$, where $T$ is the total timestep budget. Initially, once the environment models are generated via LOAM, all models are trained for the race timestep to establish baseline performance estimates. After this initial evaluation phase, training resources are allocated exclusively to the model with the highest acquisition score.

## 5 RELATED WORK

Recent studies have increasingly leveraged LLMs to improve RL by designing key components of the learning environment. Much of this work has focused on reward design, where LLMs are used to overcome sparse or delayed feedback by automatically generating reward functions (Ma et al., 2023; Kwon et al., 2023; Xie et al., 2023; Field et al., 2025). EUREKA (Ma et al., 2023) employs an evolutionary search algorithm with reflection to iteratively optimize reward model (i.e., reward function code) based on policy training statistics. Similarly, Text2Reward (Xie et al., 2023) generates dense reward model from natural language goals using a compact environment abstraction

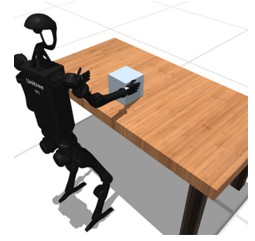 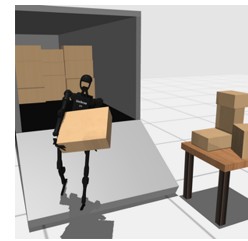

(a) Locomotion (run)      (b) Static manipulation (push)     (c) Dynamic manipulation (truck)

Figure 2: Examples of HumanoidBench tasks: (a) Locomotion: moving forward or navigating (b) Static manipulation: performing dexterous manipulation while the robot remains stationary (c) Dynamic manipulation: performing manipulation while moving.

and allows for iterative refinement via human feedback. While reward model designing has been the main emphasis, recent efforts have also begun to examine the role of observation and action models, exploring the automated specification (Chandak et al., 2019; Kim & Ha, 2021; Jia et al., 2025).

Within this direction, a recent position paper has emphasized the broader need for environment designing in RL, drawing attention to the particular difficulty of jointly designing observation and action spaces (Park et al., 2024). A closely related approach is LESR (Wang et al., 2024), which enhances RL efficiency by appending additional LLM-generated features to pre-existing, namely handcafted, observations and rewards. Unlike LESR, LOAM generates observation and action models from scratch directly from raw observation. As LLM outputs are inherently stochastic, previous works generate multiple candidates, train and evaluate them through pilot experiments iteratively, and then train on the best-performing design before final evaluation. This two-stage procedure requires additional experimentation, whereas LOAM-Race identifies promising observation–action models within a single experiment using a racing strategy.

In vision-based RL, ExploRLLM (Ma et al., 2025) utilizes Vision Language Models (VLMs) as online perception modules to construct compact, object-centric observations and actions for efficient RL training. However, while their approach focuses on visual perception primarily for pick-and-place tasks, LOAM targets a broader range of continuous control problems—including complex locomotion and manipulation—by synthesizing interfaces directly from non-visual inputs.

## 6    EXPERIMENTS

In this section, we empirically validate LOAM's capability to generate observation and action models enabling sample-efficient learning for diverse continuous control problems. Our primary evaluation is conducted on HumanoidBench (Sferrazza et al., 2024), serving as the main testbed to assess performance against baselines. To further ensure robustness and broad applicability, we extend our validation to NVIDIA Isaac Lab (Mittal et al., 2023). We first detail the experimental setup and main results on HumanoidBench, followed by the additional experiment in Isaac Lab, and conclude with ablation studies and qualitative evaluations.

### 6.1    EXPERIMENTAL SETUP

We use the HumanoidBench (Sferrazza et al., 2024) simulation environment, which is built on MuJoCo physics engine (Todorov et al., 2012), to evaluate our algorithm. It is a high-dimensional benchmark developed to accelerate algorithmic research on humanoid robots without relying on costly and fragile physical hardware. The environment features various humanoid robot models based on real-world hardware, such as the Unitree H1, equipped with dexterous hands and includes a variety of challenging whole-body manipulation and locomotion tasks as shown in Figure 2.

We evaluate our approach on 12 tasks grouped into three categories based on the required skills.

- **Locomotion**: Tasks that require the robot to move its entire body to navigate the environment. This set includes stand (maintaining a standing pose), walk (walking forward at a

speed of 1m/s), run (running forward at a speed of 5m/s), hurdle (running forward while overcoming hurdles), slide (walking forward over slides), and reach (reaching for a 3D point and touching with the left hand).

- **Static Manipulation**: Tasks that demand precise, dexterous manipulation while the robot is stationary. These are push (moving a box on a table to a target point), cube (manipulating two cubes in-hand to reach a target orientation), and insert (inserting a peg into two tight target blocks).

- **Dynamic Manipulation**: Tasks that merge locomotion and manipulation, requiring the robot to move its base while using its hands. This includes truck (unloading packages from a truck), powerlift (lifting a barbell), and package (carrying a box to a target point).

To ensure tractability, HumanoidBench provides a handcrafted $f_o^{hand}$, primarily transforming $O_{raw}$ into observations with joint positions and joint velocities and yielding a 151-dimensional feature space (or larger in some tasks [1]). We refer to this observation as $O^{f_o^{hand}}$. In contrast, we define $O_{raw}$ for HumanoidBench as the set of sensor information accessible within the MuJoCo engine, which includes not only positions and velocities but also a larger set of kinematic and dynamic data (e.g., actuator forces and body orientations represented as quaternions or rotation matrices) (See more details in Appendix D.1). This raw sensory inputs $O_{raw}$ constitutes a 3,107-dimensional space (or larger in some tasks), which is unsuitable for direct RL due to its high dimensionality. We use $O_{raw}$ only for LOAM-variants because existing methods suffer from the curse of dimensionality.

We utilize the FastTD3 as a backbone RL algorithm and OpenAI's GPT-5 (gpt-5-2025-08-07) as a backbone for all LLM-based algorithms in our experiments. FastTD3 (Seo et al., 2025) is a high-performance variant of TD3 (Fujimoto et al., 2018) tailored for humanoid control; it incorporates parallel simulation, large-batch updates, and a distributional critic to accelerate and stabilize training. For LOAM-Race, we use the number of candidates ($K$) as 3 and the race timestep as 128K. We report the average returns and standard errors over 3 random seeds.

## 6.2 Main Results

We compare LOAM and its variants against two baselines, FastTD3 and LESR. FastTD3 is an RL baseline that does not use any LLM guidance. LESR, in contrast, augments a predefined handcrafted observation, $o_{hand} \in O^{f_o^{hand}}$, using an LLM-generated function $f_o$ to produce additional features. The policy input for LESR is the concatenation $[o_{hand}, f_o(o_{hand})]$. On the other hand, LOAM only uses $f_o(o_{raw})$ as the sole input to its policy. LESR further co-generates an intrinsic reward function as an auxiliary learning signal. Importantly, LESR requires approximately 4.5× more training samples due to nine pilot experiments for best selection.[2] We also evaluate LOAM (w/ $f_o^{hand}$) to allow fair comparison under the same range of observation features.

As shown in Figure 3, LOAM and LOAM-Race consistently outperform all baselines across a wide range of tasks, achieving both higher sample-efficiency and final returns. All methods fail to learn in package, but LOAM reliably surpasses FastTD3 and LESR in the remaining tasks. Among the baselines, LESR occasionally exceeds FastTD3, but in tasks such as run, hurdle, slide, and push, its performance falls below FastTD3. This can be attributed to the high-dimensional feature space introduced by LESR, which is twice as large as the handcrafted observation, increasing observation complexity and hindering stable policy learning. Even when restricted to the same set of observation features, LOAM (w/ $f_o^{hand}$) outperforms LESR across most tasks, indicating that its advantage does not stem from simply having more sensory inputs. This difference is explained by the design approach: LESR appends extra features to handcrafted observations, whereas LOAM, instead of augmenting what may be a suboptimal handcrafted observation, directly designs a compact and task-relevant one.

While LOAM demonstrates strong sample-efficiency and quickly reaches near-optimal performance, it exhibits instability in certain tasks such as insert_small and cube, where the LLM often generates observation and action models that lack informative features. LOAM-Race, on the other hand,

---

[1] For instance, in h1hand-reach-v0, the 3D positions of left hand and target goal are additionally provided.

[2] LESR runs nine pilot experiments (three iterations of generating three candidates, each trained with half the total budget), after which the best observation design is used for the final report.

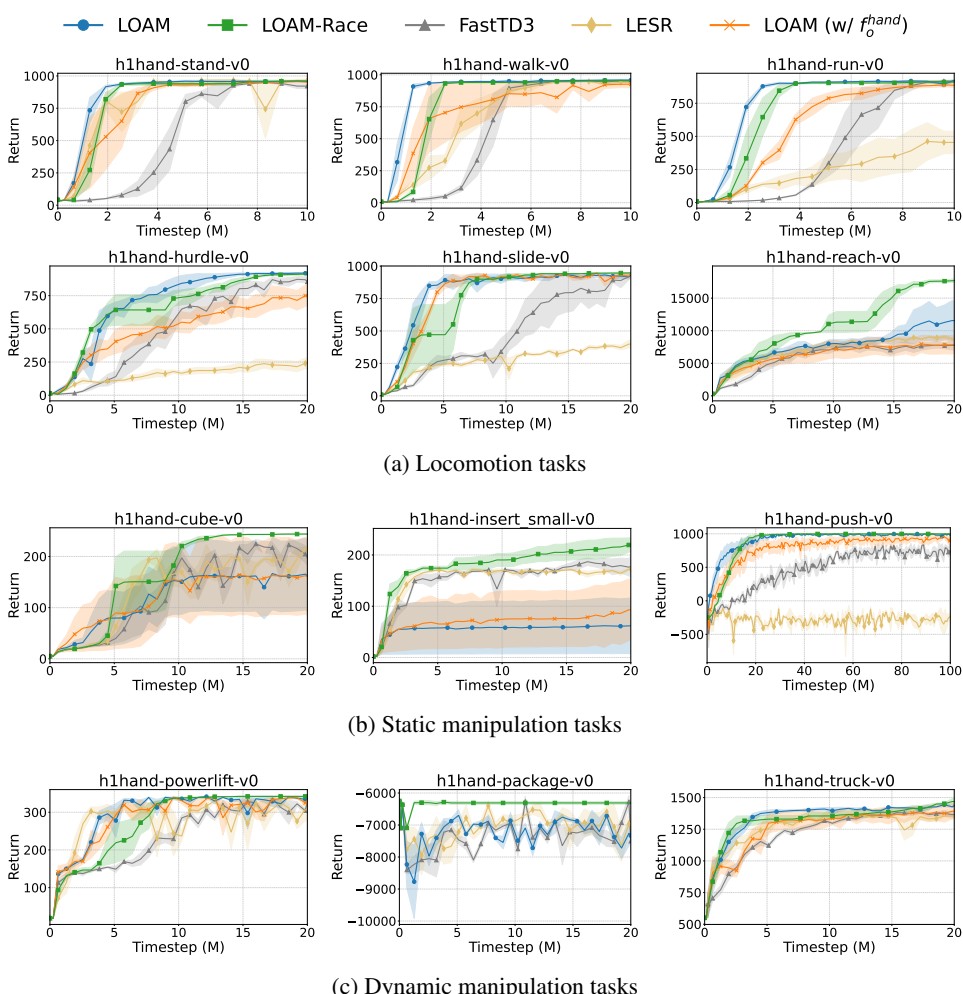

(a) Locomotion tasks

(b) Static manipulation tasks

(c) Dynamic manipulation tasks

Figure 3: Learning curves on 12 HumanoidBench tasks. We report the average return over 3 random seeds, with shaded regions indicating standard error.

slightly decreases sample-efficiency compared to LOAM, but it converges to the highest asymptotic returns across all tasks. This improvement indicates that LOAM-Race effectively enhances robustness against the inherent randomness in LLM-generated designs.

A particularly notable case is the reach task, where a return of 12,000 points serves as the success threshold. None of the prior baselines, including FastTD3 and LESR, were able to achieve, failing to solve the task. In contrast, both LOAM and LOAM-Race surpass the 12,000 points threshold and achieve substantially higher final returns. In particular, LOAM-Race attains the highest return with minimal variance within the training budget. This performance gain is attributed to the LOAM's ability to generate compact and task-relevant observation and action spaces, which facilitates more efficient exploration. A qualitative analysis of the LOAM-designed models, including the complete code and line-by-line descriptions, can be found in Appendix B.

## 6.3 ADDITIONAL EVALUATION ON ISAAC LAB

We extended our evaluation to NVIDIA Isaac Lab (Mittal et al., 2023) to demonstrate LOAM's broad applicability beyond the primary HumanoidBench benchmark. This domain employs the PhysX simulation engine (NVIDIA, 2020), presenting a distinct environmental setting compared to the HumanoidBench used in our main experiments. We select three representative tasks: Lift-Cube and Open-Drawer (Franka Emika Panda), and Repose-Cube (Allegro Hand).

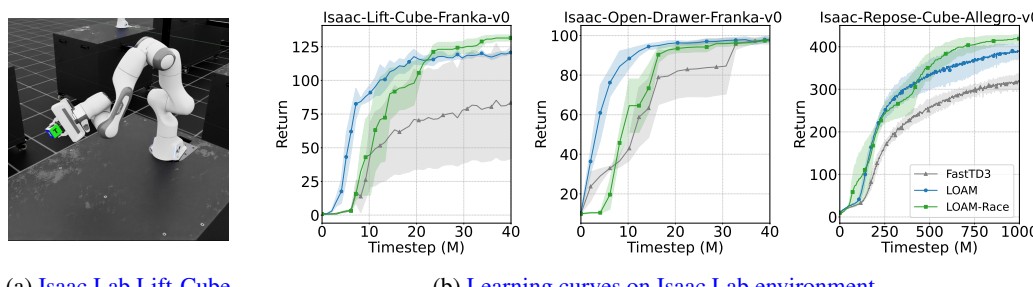

(a) Isaac Lab Lift-Cube

(b) Learning curves on Isaac Lab environment.

Figure 4: Performance on Isaac Lab environment. (a) Picking a cube and bring it to a target position with the Franka robot. (b) Learning curves on three tasks showing both LOAM and LOAM-Race enable sample-efficient learning compared to FastTD3.

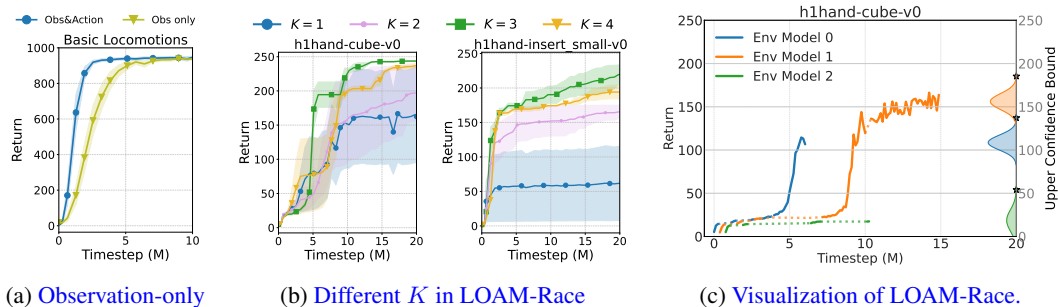

(a) Observation-only        (b) Different $K$ in LOAM-Race        (c) Visualization of LOAM-Race.

Figure 5: Further experiments on LOAM and its variants: (a) LOAM with observation-only design, (b) Impact of the number of candidates $(K)$, and (c) The left and right y-axes denote the candidate's average return and the acquisition score, respectively.

To adapt LOAM to the Isaac Lab environment, we substitute domain-specific details from official specifications[3] into the corresponding components such as *Task Description* within our structured prompts (Table 1). We believe this modular design demonstrates that LOAM can be extended to other domains simply by providing the relevant domain-specific context.

Similar to the HumanoidBench setup, Isaac Lab provides handcrafted observation models yielding compact feature spaces (exceeding 30 dimensions per task). In contrast, we define the raw observation space $(O_{raw})$ as the set of sensor information accessible through `isaaclab.assets.RigidObjectData` (see Appendix D.2 for the complete list)[4]. Notably, these raw sensory inputs constitute a high-dimensional space exceeding 1,100 dimensions for each task. See Appendix E for further details.

As illustrated in Figure 4, LOAM successfully generalizes to this new environment, significantly outperforming the FastTD3 baseline across all tasks. These results confirm that our framework is not overfitting to a specific benchmark or simulation engine but is robust capable of autonomously discovering effective policies.

### 6.4 FURTHER ANALYSIS

**Effect of the action design.** We compare the full LOAM framework against its observation-only variant on locomotion tasks, with results shown in Figure 5a. Basic locomotion tasks (stand, walk, run) benefit from action design by reaching peak performance faster. This indicates that observation design alone improves performance, but combining it with action design maximizes the effect.

---

[3]Please refer to https://isaac-sim.github.io/IsaacLab/main/index.html for details.

[4]To compel the framework to autonomously discover necessary information directly from raw data, we intentionally excluded highly-engineered features provided in Isaac Lab.

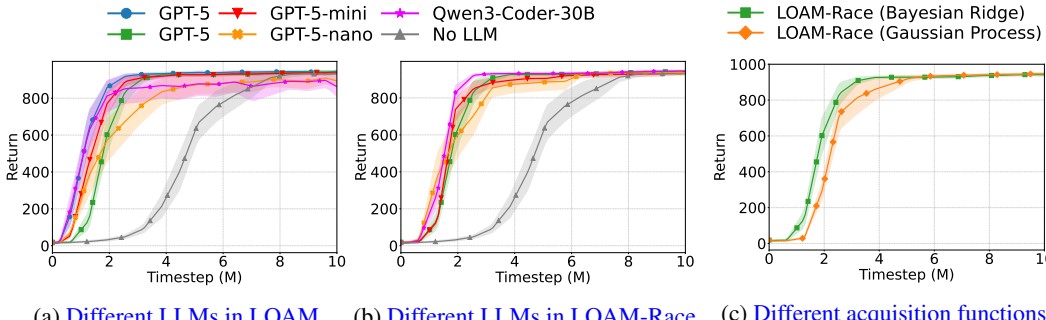

(a) Different LLMs in LOAM    (b) Different LLMs in LOAM-Race    (c) Different acquisition functions

Figure 6: Further experiments on HumanoidBench the basic locomotion tasks. (a-b) Comparison of different LLMs including GPT-5 variants, Qwen3-Coder-30B, and a baseline without LLM (FastTD3), (c) Comparison of LOAM-Race with Bayesian Ridge versus Gaussian Process.

**Effect of the number of candidates in LOAM-Race.** Figure 5b illustrates the trade-off between robustness and convergence speed. With more candidates, performance variance narrows, consistent with the intuition that a larger pool is more likely to include at least one strong candidate and thereby improves robustness. However, once a strong candidate is present, additional ones merely dilute the training budget, slowing convergence, as seen in $K = 4$ being slower than $K = 3$. From this ablation, we find that using $K = 3$ provides a favorable balance between robustness and efficiency.

**Visualization of LOAM-Race.** We visualize the racing procedure of LOAM-Race during training in Figure 5c. Every race timesteps (128K timesteps), LOAM-Race estimates the upper confidence bounds of all $K$ candidate environment models and selects one accordingly. The following race timesteps are then used to update the policy associated with the selected models, and subsequently the corresponding upper confidence bound is further refined. In this figure, LOAM-Race concentrates resources on promising candidates (models 0 and 1) while, after minimal exploration, allocating essentially none to the weak candidate (model 2).

**Effect of the LLM backbone.** Our ablation study comparing the standard GPT-5 backend against lighter and open-weights variants—GPT-5-mini, GPT-5-nano, and Qwen3-Coder-30B-A3B-Instruct—reveals that while standard LOAM exhibits performance degradation with reduced model capacity (particularly in 30B and nano models) (Figure 6a), LOAM-Race effectively mitigates this sensitivity (Figure 6b). Notably, LOAM-Race enables these smaller models to achieve results comparable to the GPT-5, confirming that our racing mechanism unlocks the potential of cost-effective backbones and reduces dependency on proprietary systems (See more details in Appendix F.1).

**Effect of the acquisition function.** To validate our choice of Bayesian Ridge Regression (BRR), we compared it against a Gaussian Process Regression (GPR) capable of non-linear modeling. The results indicate that the increased complexity of GPR offers no significant advantage, as both functions yield nearly identical asymptotic returns. In fact, BRR exhibited slightly faster convergence and improved stability (See more details in Appendix F.2).

## 7 CONCLUSION

We presented LOAM, a framework that automatically designs observation and action models for RL agents, and LOAM-Race, a competitive racing mechanism that handles LLM output variability through efficient parallel evaluation. Our experiments show that this automated approach not only substantially reduces the need for manual feature engineering but also outperforms human-designed models across all HumanoidBench tasks. By automating the design of observation and action models, we alleviate a critical bottleneck that has limited RL deployment in complex systems. Our current framework focuses on automating observation and action model design, acting as a complementary approach to automatic reward design. While reward design shapes the learning objective, observation-action design determines the interface through which the agent perceives and interacts with the environment. Consequently, integrating LOAM with reward design methods could further reduce human intervention in RL systems, presenting an exciting future direction.

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

## A   FULL PROMPT TEMPLATES AND DESIGN RATIONALE

This section details the structured full prompts that drive the LOAM framework. Our methodology is a modular pipeline that begins with a **System Prompt** to provide task-specific context and define the LLM's expert persona. It then deploys its core modules for **Observation Model Design**, which generates a compact and informative state representation for the agent, and **Action Model Design**, which structures a coordinated and efficient control interface for the robot's actuators.

To ground these prompts in a concrete example, the templates shown are tailored for the H1Hand robot. The H1Hand morphology consists of a floating pelvis, two five-DoF legs, a single-DoF torso joint, two five-DoF arms, and two dexterous five-digit hands. The simulator exports 76 generalized positions (qpos) and 75 generalized velocities (qvel). This detailed structural information, including index mappings for joints and sensors, is provided to the LLM within the System Prompt to ensure any generated code can address the correct components.

Figure 7 provides a visual overview of the stages involved in this pipeline.

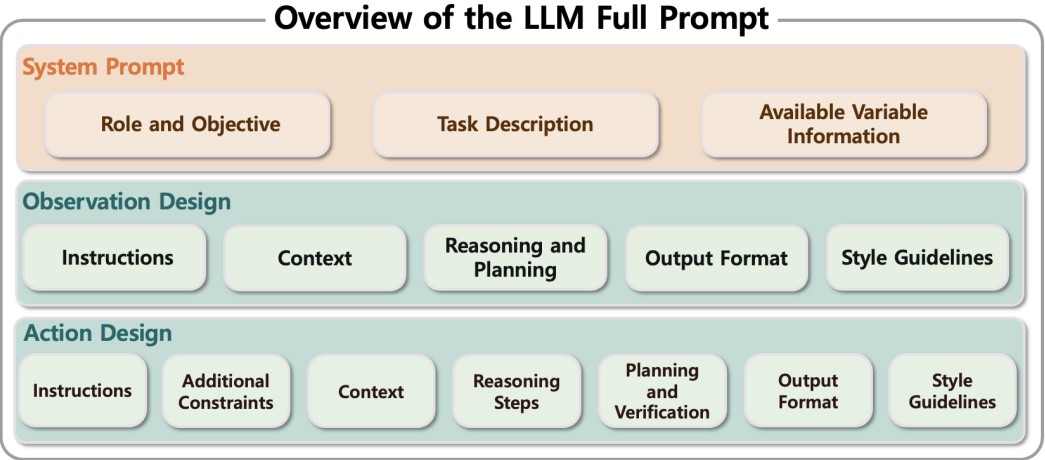

Figure 7: Overview of the LLM Full Prompt Template. The template is organized into three main components. System Prompt establishes the LLM's role, the task Description, and the Available Variable Info. Observation Model Design details the instructions, context, reasoning, output format, and style for generating the state representation function. Action Model Design provides a similar breakdown for synthesizing the action space, including additional constraints, reasoning steps for design, and verification plans.

### A.1   PROMPT SUITE RATIONALE AND STRUCTURE

Our framework is built upon a structured suite of prompts, where each module serves a distinct function in the code generation pipeline. As illustrated in Figure 7, this suite is composed of three primary modules. Below, we detail the rationale and full template for each component.

- **System Prompt:** The pipeline begins with the System Prompt, which grounds the LLM in the global context. it serves three primary functions: defining the LLM's expert persona **Role and Objective**, providing the complete operational details of the task **Task Description** (Context), and detailing all available simulator state variables **Available Variable Information** (Data attributes). This initial stage ensures the LLM is fully contextualized before any code generation begins.

756
757
758
759
760
761
762
763
764
765
766
767
768
769
770
771
772
773
774
775
776
777
778
779
780
781
782
783
784
785
786
787
788
789
790
791
792
793
794
795
796
797
798
799
800
801
802
803
804
805
806
807
808
809

**System Prompt**

```
# Role and Objective
You are an expert robotics engineer and AI researcher specializing in Deep
 Reinforcement Learning. Your objective is to design an optimal, {TYPES}
for a reinforcement learning agent, enabling efficient and effective
learning for the specified task.

# Context
- Task description :
{TASK_DESCRIPTION}
- The `data` object holds simulation states; all attributes are torch
tensors with a batch dimension as the first axis.
- Detailed information about the "data" object's attributes can be found
in the below " Detailed data attributes" section, which describes what
each index in the data attribute represents.

# Data attributes
{AVAILABLE_STATE_INFO}
```

- **Observation Model Design Prompt:** This module guides the synthesis of a self-contained Python function, compute_obs. The prompt is structured with five key components. It provides explicit **Instructions** on the function's requirements and defines its operational **Context**, such as its signature. A **Reasoning and Planning** section prompts the LLM to select and engineer relevant features. Finally, the prompt enforces a specific **Output Format** (a single code block) and sets **Style Guidelines** (Verbosity), such as code verbosity and the use of vectorized operations, to ensure the generated code is efficient and robust for RL training.

**Observation Model Design Prompt**

```
# Instructions
- Begin with a concise checklist (3-7 bullets) of what you will do; keep
items conceptual, not implementation-level.
- **Generate a complete Python function** named `compute_obs` that
computes **and returns** the observation vector named `obs` (as a torch
tensor).
- The function signature **must be** `compute_obs(data)`. It should accept
 the simulation state object `data` as its sole argument.
- The function **must return** the final `obs` tensor.
- Inside the function, use only the explicitly allowed state variables
from the `data` object. However, never use cfrc_ext data for any external,
 non-robot objects.
- If you need a value not in the allowed variables, hard-code it as a
constant per the Robot Specification-never try to access `data` or `self`
for unspecified attributes.
- The function's logic must correctly handle the batch dimension and be
fully vectorized.
- Any new tensors created inside the function must use the same device as
existing tensors (e.g., `device=data.qpos.device`).
- The output must be a single, raw Python code block labeled `python`,
containing the entire function. Do not include any other text or
explanations outside the code block.

# Context
- **Function Signature:** `def compute_obs(data):`
- **Inputs:** The function will receive `data` (simulation state and
sensor data).
```

```
# Reasoning and Planning
- **Plan the complete function structure**, including the signature `
compute_obs(data)`, the internal logic for vector computation, and the
final `return obs` statement.
- Internally: Analyze variables and index mappings. Implement device-safe,
 batch-capable extraction and concatenation with the correct shapes.
- **Considering the task description carefully**, select the most relevant
 features (joint positions/velocities, site positions, target position)
for the task. If necessary, leverage your domain expertise in robotics and
 physics to compute derived, physically meaningful features.
- For any physical task, maintaining the robot's stable and balanced
posture is a fundamental and implicit requirement. The final code must
reflect a deep understanding of the need for whole-body postural control
while performing the primary task.
- After generating the function, validate that the returned observation
vector uses only allowed variables and that batch/device compatibility is
preserved. If validation fails, correct and re-verify.

# Output Format
- Output a single markdown `python` code block.
- The code block must begin with a detailed comment explaining the
observation design and structure.
- The code block must contain the **entire, self-contained `compute_obs`
function**, from the `def` statement to the final `return` statement.
- Do not include any inline/trailing comments or any text outside the code
 block.

# Verbosity
- Write code with high clarity and a readable structure. Be verbose in
comments at the top of the code block only.

Write a **complete and self-contained Python function** named `compute_obs
` that computes and returns the effective observation vector for efficient
 training of a deep reinforcement learning agent in this environment.
```

- **Action Model Design Prompt:** This prompt addresses the high-dimensional control challenge by instructing the LLM to design an *Action Expansion*: a deterministic mapping from a low-dimensional latent action to the full 61-actuator command vector. This mapping is typically achieved via predefined synergies—coordinated patterns where a single latent command drives multiple joints—and stabilizing PD terms. The prompt requires the LLM to first reason about an optimal latent action space based on the task and robot morphology.

To structure this process, the prompt provides detailed **Instructions**, defines the function **Context** including the full actuator map, and specifies **Additional Constraints** such as shape validation. It then guides the LLM through **Reasoning Steps** for synergy design and a **Planning and Verification** stage to ensure a correct mapping to the full actuator space. A strict **Output Format** and specific **Style Guidelines** are enforced to produce an efficient compute_action function that embeds strong domain priors, thereby constraining the agent's exploration problem.

## Action Model Design Prompt

```
# Instructions
- Begin with a concise checklist (3-7 bullets) of what you will do.
- **First, determine and state an optimal, dimension for the input action
space (`action_dim`)** based on the actuator map and robotics principles.
- **Generate a complete Python function** with the signature `
compute_action(action, data)`.
```

```
- The function must accept a `action` tensor and the `data` object as
inputs.
- The function must transform the `action` tensor (with the `action_dim`
you determined) into a full 61-dimensional control vector.
- The function **must return** the final 61-dimensional tensor, named `
action_out`.
- **Your output must follow the two-part format specified in the "Output
Format" section below.**
- The Python code block must start with a high-level comment summarizing
your chosen `action_dim` and mapping approach.
- All implementation must accommodate batch processing and handle device
consistency correctly.

## Sub-categories
- All tensor shapes must be validated to prevent mismatches.
- Any new tensors must be created on the same device as the input tensor `
action`.
- Do not rename the function name or its parameters.

# Context
- **Function Signature:** `def compute_action(action, data):`
- **Inputs:**
    - `action` (torch.Tensor): **The input action tensor is the direct
output from a policy neural network. Its values are typically in the range
 [-1, 1], resulting from an activation function like `tanh`.**
    - `data` (simulation state and sensor data)
- Actuator mapping with 61 total outputs detailed below

# Actuator mapping (nu=61):
The final 61-dimensional output vector corresponds to the following
actuators. **Use this detailed map to decide on an appropriate input
dimension and how it should be expanded.**
Part 1: Body and Limbs (Indices 0-20)
# Legs (0-9)
[0] left_hip_yaw: Left hip's side-to-side rotation (Yaw)
[1] left_hip_roll: Left hip's side-to-side tilt (Roll)
[2] left_hip_pitch: Left hip's front-to-back movement (Pitch)
[3] left_knee: Left knee bending
[4] left_ankle: Left ankle movement
[5] right_hip_yaw: Right hip's side-to-side rotation (Yaw)
[6] right_hip_roll: Right hip's side-to-side tilt (Roll)
[7] right_hip_pitch: Right hip's front-to-back movement (Pitch)
[8] right_knee: Right knee bending
[9] right_ankle: Right ankle movement
# Torso (10)
[10] torso: Torso side-to-side rotation
# Arms (11-20)
[11] left_shoulder_pitch: Left shoulder's front-to-back movement (Pitch)
[12] left_shoulder_roll: Left shoulder's up-and-down movement (Roll)
[13] left_shoulder_yaw: Left shoulder's side-to-side rotation (Yaw)
[14] left_elbow: Left elbow bending
[15] left_wrist_yaw: Left wrist's side-to-side rotation (Yaw)
[16] right_shoulder_pitch: Right shoulder's front-to-back movement (Pitch)
[17] right_shoulder_roll: Right shoulder's up-and-down movement (Roll)
[18] right_shoulder_yaw: Right shoulder's side-to-side rotation (Yaw)
[19] right_elbow: Right elbow bending
[20] right_wrist_yaw: Right wrist's side-to-side rotation (Yaw)
Part 2: Left Hand (Indices 21-40)
# Wrist (21-22)
[21] lh_A_WRJ2: Left wrist pitch (up/down movement)
[22] lh_A_WRJ1: Left wrist roll (side-to-side tilt)
# Thumb (23-27)
```

```
[23] lh_A_THJ5: Left thumb base joint rotation
[24] lh_A_THJ4: Left thumb proximal joint (first knuckle) bend
[25] lh_A_THJ3: Left thumb hub joint
[26] lh_A_THJ2: Left thumb middle joint
[27] lh_A_THJ1: Left thumb distal joint (tip)
# Fingers (28-40)
[28] lh_A_FFJ4: Left index finger knuckle (side-to-side)
[29] lh_A_FFJ3: Left index finger proximal bend
[30] lh_A_FFJ0: Left index finger middle/distal bend (tendon)
[31] lh_A_MFJ4: Left middle finger knuckle (side-to-side)
[32] lh_A_MFJ3: Left middle finger proximal bend
[33] lh_A_MFJ0: Left middle finger middle/distal bend (tendon)
[34] lh_A_RFJ4: Left ring finger knuckle (side-to-side)
[35] lh_A_RFJ3: Left ring finger proximal bend
[36] lh_A_RFJ0: Left ring finger middle/distal bend (tendon)
[37] lh_A_LFJ5: Left little finger metacarpal joint
[38] lh_A_LFJ4: Left little finger knuckle (side-to-side)
[39] lh_A_LFJ3: Left little finger proximal bend
[40] lh_A_LFJ0: Left little finger middle/distal bend (tendon)
Part 3: Right Hand (Indices 41-60)
# This part mirrors the left hand's structure
# Wrist (41-42), Thumb (43-47), Fingers (48-60)
[41] rh_A_WRJ2: Right wrist pitch (up/down movement)
...
[60] rh_A_LFJ0: Right little finger middle/distal bend (tendon)

# Reasoning Steps
- **Plan the complete function structure**, including the signature `
compute_action(action, data)`, the internal mapping logic, and the final `
return action_out` statement.
- **Propose a suitable `action_dim`. ** Justify this choice in the code's
summary comment.
- **Considering the task description carefully**, transform the action
tensor intelligently to the full 61-dimensional space to create a smooth
control landscape for the neural network.
- For any physical task, maintaining the robot's stable and balanced
posture is a fundamental and implicit requirement. The final code must
reflect a deep understanding of the need for whole-body postural control
while performing the primary task.
- Ensure correct expansion respecting batch and device.

# Planning and Verification
- Carefully decompose the expansion from **your chosen `action_dim`** to
61 using the context and actuator map.
- Verify shape compatibility for all operations and ensure tensor device
consistency.
- After generating `action_out` but before returning it, include in-code
validation to confirm the output's batch and dimensional shape.

# Output Format
- **Your output must consist of two distinct parts, in this specific order
. Do not include any other text or explanation.**
- **Part 1: Action Dimension**
    - First, output the identifier `dimension`, followed by a newline, and
 then the integer value for the chosen action dimension.
    - Example:
    ```
    dimension
    20
    ```
- **Part 2: Python Code**
```

```
    - Following the dimension, output a single Python code block with the
language identifier `python`.
    - This block must contain the **entire, self-contained `compute_action
` function**, from the `def` statement to the final `return` statement.

# Verbosity
- Code must include clear structure. Be verbose in the summary comment at
the top of the code block only.

First, determine and output an optimal action space dimension. Then, write
 a **complete and self-contained Python function** named `compute_action`
that expands an action tensor from this dimension into the full 61-
dimensional control vector and **returns** the result.
"""
```

## B  QUALITATIVE ANALYSIS OF LOAM-DESIGNED OBSERVATION AND ACTION MODELS

In this section, we provide a qualitative analysis of the LOAM-Designed observation and action models. We use the `h1hand-run-v0` locomotion task as an in-depth case study. The analysis first presents the fully populated prompt provided to the LLM for this task. It then details the baseline model, followed by the full code generated by our framework. Finally, we examine the generated design to identify the specific choices and structures that likely contribute to its effective performance.

### B.1  EXAMPLE OF A POPULATED PROMPT FOR `h1hand-run-v0`

The placeholders within the `System Prompt` templates shown in Section A, such as {TYPES}, {TASK_DESCRIPTION}, and {AVAILABLE_STATE_INFO}, are dynamically populated with the specifics of each target environment. This process grounds the LLM in the precise context of the task before code generation. For any given generation step, the system prompt templates are populated and concatenated following each specific tasks. For instance, to generate the designs for `h1hand-run-v0`, the system first populates the `System Prompt` template with the task's specific details and then appends the two universal templates for `Observation Model Design` and `Action Model Design` that are detailed in Section A. As a concrete example, the populated System Prompt for h1hand-run-v0 is shown below.

---

**Example: System Prompt for `h1hand-run-v0`**

```
# --- Start of Populated System Prompt (from System Prompt Template) ---
# Role and Objective
You are an expert robotics engineer and AI researcher specializing in Deep
Reinforcement Learning. Your objective is to design an optimal, observation
function code, for a reinforcement learning agent, enabling efficient and
effective learning for the specified task.

# Context
- Task description :
Name: h1hand-run-v0
Objective: Keep forward velocity close to 5 m/s without falling to the ground.
Initialization: The robot is initialized to a standing position, with random noise
added to all joint positions during each episode reset.
Termination: The episode terminates after 1000 steps, or when z_pelvis < 0.2.

- The `data` object holds simulation states; all attributes are torch tensors with
 a batch dimension as the first axis.
- Detailed information about the "data" object's attributes can be found in the
below " Detailed data attributes" section, which describes what each index in the
data attribute represents.

# Data attributes
data.qpos
 - Shape: (batch, nq=76)
 - Description: Generalized positions (joint angles, base pose).
data.qvel
 - Shape: (batch, nv=75)
 - Description: Generalized velocities (joint rates, base velocity).
data.site_xpos
 - Shape: (batch,nsite=9, 3)
 - Description: 3D position of each defined site (useful for end-effectors,
sensors, etc.).
data.qfrc_actuator
 - Shape: (batch, nv=75)
 - Description: Actuator forces mapped to degrees of freedom.
```

```
data.xpos
 - Shape: (batch, nbody=71, 3)
 - Description: 3D position of each body.
data.xquat
 Shape: (batch, nbody=71, 4)
 Description: 4D orientation (quaternion) of each body.
data.cvel
 Shape: (batch, nbody=71, 6)
 Description: 6D spatial velocity (linear & angular) of each body.
data.cfrc_ext
 Shape: (batch, nbody=71, 6)
 Description: 6D external wrench (force & torque) on each body.
data.cinert
 Shape: (batch,nbody=71, 10)
 Description: 10D composite rigid body inertia of each body.
data.actuator_force
 Shape: (batch, nu=61)
 Description: Force/torque generated by each actuator.
data.sensordata
 Shape: (batch, nsensor=14)
 Description: Scalar sensor outputs.

Index Mapping for Key Arrays
This section provides the name-to-index mapping for the primary arrays.

data.qpos (nq=76, Generalized Positions)
[0:7] free_base: Pelvis pose ([x, y, z, qw, qx, qy, qz])
[7:12] left_leg: [hip_yaw, hip_roll, hip_pitch, knee, ankle]
[12:17] right_leg: [hip_yaw, hip_roll, hip_pitch, knee, ankle]
[17] torso
[18:23] left_arm: [shoulder_pitch, shoulder_roll, shoulder_yaw, elbow, wrist_yaw]
[23:47] left_hand: 24 joints
 - [23:25] Wrist (2j)
 - [25:29] Index Finger (4j)
 - [29:33] Middle Finger (4j)
 - [33:37] Ring Finger (4j)
 - [37:42] Little Finger (5j)
 - [42:47] Thumb (5j)
[47:52] right_arm: [shoulder_pitch, shoulder_roll, shoulder_yaw, elbow, wrist_yaw]
[52:76] right_hand: 24 joints
 - [52:54] Wrist (2j)
 - [54:58] Index Finger (4j)
 - [58:62] Middle Finger (4j)
 - [62:66] Ring Finger (4j)
 - [66:71] Little Finger (5j)
 - [71:76] Thumb (5j)

data.qvel (nv=75, Generalized Velocities)
[0:6] free_base: Pelvis velocity ([vx, vy, vz, wx, wy, wz])
[6:11] left_leg: 5 joint velocities
[11:16] right_leg: 5 joint velocities
[16] torso: 1 joint velocity
[17:22] left_arm: 5 joint velocities
[22:46] left_hand: 24 joint velocities (same structure as qpos)
[46:51] right_arm: 5 joint velocities
[51:75] right_hand: 24 joint velocities (same structure as qpos)

data.site_xpos (nsite=9, Site Positions)
[0] com
[1] left_foot
```

```
    [2] right_foot
    [3] left_eye
    [4] right_eye
    [5] head
    [6] imu
    [7] left_hand
    [8] right_hand

    data.xpos, data.xquat, data.cvel, data.cfrc_ext, data.cinert (nbody=71,
    Body-related Arrays)
    [0] world
    [1] pelvis
    [2:7] left_leg_links: [hip_yaw, hip_roll, hip_pitch, knee, ankle]
    [7:12] right_leg_links: [hip_yaw, hip_roll, hip_pitch, knee, ankle]
    [12] torso_link
    [13:17] left_arm_links: [shoulder_pitch, shoulder_roll, shoulder_yaw, elbow]
    [17:42] left_hand: 25 bodies
     [17:20] Left Palm area (left_hand, wrist, palm)
     [20:24] Left Index Finger (knuckle, proximal, middle, distal)
     [24:28] Left Middle Finger (4 bodies)
     [28:32] Left Ring Finger (4 bodies)
     [32:37] Left Little Finger (5 bodies)
     [37:42] Left Thumb (5 bodies)
    [46:71] right_hand: 25 bodies (same structure as left_hand)

    data.actuator_force (nu=61, Actuator Forces)
    [0:5] left_leg: 5 actuators
    [5:10] right_leg: 5 actuators
    [10] torso: 1 actuator
    [11:16] left_arm: 5 actuators
    [16:21] right_arm: 5 actuators
    [21:41] left_hand: 20 actuators
     [21:23] Left Wrist (2a)
     [23:28] Left Thumb (5a)
     [28:31] Left Index Finger (3a)
     [31:34] Left Middle Finger (3a)
     [34:37] Left Ring Finger (3a)
     [37:41] Left Little Finger (4a)
    [41:61] right_hand: 20 actuators
     [41:43] Right Wrist (2a)
     [43:48] Right Thumb (5a)
     [48:51] Right Index Finger (3a)
     [51:54] Right Middle Finger (3a)
     [54:57] Right Ring Finger (3a)
     [57:61] Right Little Finger (4a)

    data.sensordata (nsensor=14, Sensor Outputs)
    [0] left_foot_sensor (1D touch)
    [1] right_foot_sensor (1D touch)
    [2:5] pelvis_subtreelinvel (3D velocity)
    [5:8] left_hand_subtreelinvel (3D velocity)
    [8:11] right_hand_subtreelinvel (3D velocity)
    [11:14] body_velocimeter (3D velocity)
```

**Observation Model Design Prompt**

The content of this prompt is identical to the "Observation Model Design" template shown in Section A.1.

**Action Model Design Prompt**

The content of this prompt is identical to the "Action Model Design" template shown in Section A.1.

## B.2    CASE STUDY 1: h1hand-run-v0 (SUCCESSFUL LOCOMOTION)

The objective of this task is to command the H1 humanoid robot to run forward at a target velocity of 5.0 m/s while maintaining balance. This task primarily tests high-speed locomotion, dynamic stability, and rhythmic coordination.

### B.2.1    BASELINE MODEL ANALYSIS

The baseline model utilizes a standard configuration where the observation and action spaces are high-dimensional and minimally processed.

- **Observation Space (151 dimensions):** The observation vector is formed by the direct concatenation of the robot's generalized positions (qpos, 76 dimensions) and velocities (qvel, 75 dimensions). This representation is comprehensive but includes task-irrelevant information (e.g., finger joint states) and is sensitive to the agent's global heading.
- **Action Space (61 dimensions):** The policy directly outputs a 61-dimensional vector, where each dimension corresponds to one of the robot's actuators. This provides full control but forces the agent to learn all inter-joint coordinations from scratch.

### B.2.2    FULL LOAM-DESIGNED MODELS CODE FOR h1hand-run-v0

**LOAM-Designed Observation Model for `h1hand-run-v0`**

```
1   # Observation design for h1hand-run-v0 (final, aligned with 13D action)
2   # Goal: Run forward at ~5 m/s without falling. Observations emphasize base
        state in a heading-invariant manner,
3   # task goal, contacts, feet geometry, and only the joint subset that the
        policy controls (or influences via synergy).
4   #
5   # Structure (all batched torch tensors):
6   # 1) Base height:
7   #    - pelvis_z: qpos[:, 2] (1)
8   # 2) Base orientation/context (yaw invariant):
9   #    - gravity_in_base (3): world gravity [0,0,-1] expressed in pelvis frame
        via quaternion rotation
10  #    - heading_xy (2): pelvis forward (+X in pelvis) projected to world XY,
        normalized -> [cos(yaw), sin(yaw)]
11  # 3) Base velocities (in pelvis frame):
12  #    - v_body (3): linear velocity from sensordata (body_velocimeter)
13  #    - w_body (3): base angular velocity (qvel[:,3:6]) rotated into pelvis
        frame
14  # 4) Task-specific signals:
15  #    - speed_target (1): constant 5.0 m/s
```

```python
16  #      - speed_error (1): v_body[:, 0] - 5.0 (forward speed error in pelvis
        #        frame)
17  # 5) Contacts:
18  #      - foot_contact (2): [left_foot_sensor, right_foot_sensor]
19  # 6) Geometry:
20  #      - feet_pos_in_base (6): left/right foot site positions relative to
        #        pelvis, expressed in pelvis frame
21  # 7) Controlled joint subset states (positions and velocities):
22  #      - pos (13): [L hip_roll, L hip_pitch, L knee, L ankle,
23  #                   R hip_roll, R hip_pitch, R knee, R ankle,
24  #                   torso,
25  #                   L shoulder_pitch, L elbow,
26  #                   R shoulder_pitch, R elbow]
27  #      - vel (13): same ordering as pos
28  #
29  # Total observation dimension: 1 + 3 + 2 + 3 + 3 + 1 + 1 + 2 + 6 + 13 + 13 =
        48
30  def compute_obs(data):
31      import torch
32
33      device = data.qpos.device
34      dtype = data.qpos.dtype
35
36      # Quaternion helpers (q = [w, x, y, z])
37      def quat_normalize(q):
38          return q / torch.clamp(torch.linalg.norm(q, dim=-1, keepdim=True),
        min=1e-8)
39
40      def quat_conj(q):
41          qc = q.clone()
42          qc[..., 1:] = -qc[..., 1:]
43          return qc
44
45      # Rotate vector(s) v by quaternion q (unit). Supports v shape (B,3) or (B
        ,N,3).
46      def quat_rotate(q, v):
47          if v.dim() == q.dim() + 1:
48              q = q.unsqueeze(-2).expand(*v.shape[:-1], 4)
49          q = quat_normalize(q)
50          qvec = q[..., 1:]        # (..., 3)
51          qw = q[..., :1]          # (..., 1)
52          t = 2.0 * torch.cross(qvec, v, dim=-1)
53          return v + qw * t + torch.cross(qvec, t, dim=-1)
54
55      B = data.qpos.shape[0]
56
57      # Base pose
58      base_pos = data.qpos[:, 0:3]                # (B,3) world
59      base_quat = quat_normalize(data.qpos[:, 3:7])  # (B,4)
60      base_quat_conj = quat_conj(base_quat)
61
62      # 1) Base height
63      pelvis_z = base_pos[:, 2:3] # (B,1)
64
65      # 2) Base orientation/context
66      g_world = torch.tensor([0.0, 0.0, -1.0], device=device, dtype=dtype).
        expand(B, 3)
67      gravity_in_base = quat_rotate(base_quat_conj, g_world)  # (B,3)
68
69      ex_body = torch.tensor([1.0, 0.0, 0.0], device=device, dtype=dtype).
        expand(B, 3)
70      fwd_world = quat_rotate(base_quat, ex_body)  # (B,3)
```

```
71        heading_xy = fwd_world[:, :2]
72        heading_norm = torch.clamp(torch.linalg.norm(heading_xy, dim=-1, keepdim=
           True), min=1e-8)
73        heading_xy = heading_xy / heading_norm  # (B,2)
74
75        # 3) Base velocities (expressed in pelvis frame)
76        v_body = data.sensordata[:, 11:14]  # (B,3) already in body frame
77        w_world = data.qvel[:, 3:6]         # (B,3) world frame
78        w_body = quat_rotate(base_quat_conj, w_world)  # (B,3)
79
80        # 4) Task-specific signals
81        speed_target = torch.full((B, 1), 5.0, device=device, dtype=dtype)
82        speed_error = v_body[:, 0:1] - speed_target  # (B,1)
83
84        # 5) Contacts
85        foot_contact = data.sensordata[:, 0:2]  # (B,2)
86
87        # 6) Feet positions relative to pelvis, in pelvis frame
88        left_foot_world = data.site_xpos[:, 1, :]   # (B,3)
89        right_foot_world = data.site_xpos[:, 2, :]  # (B,3)
90        lf_rel_world = left_foot_world - base_pos
91        rf_rel_world = right_foot_world - base_pos
92        lf_in_base = quat_rotate(base_quat_conj, lf_rel_world)  # (B,3)
93        rf_in_base = quat_rotate(base_quat_conj, rf_rel_world)  # (B,3)
94
95        # 7) Controlled joint subset (positions and velocities)
96        # qpos indices
97        qpos_idx = [
98            8, 9, 10, 11,     # L leg: hip_roll, hip_pitch, knee, ankle
99            13, 14, 15, 16,   # R leg: hip_roll, hip_pitch, knee, ankle
100           17,               # torso
101           18, 21,           # L arm: shoulder_pitch, elbow
102           47, 50,           # R arm: shoulder_pitch, elbow
103       ]
104       joint_pos = data.qpos[:, qpos_idx]  # (B,13)
105
106       # qvel indices
107       qvel_idx = [
108           7, 8, 9, 10,      # L leg: hip_roll, hip_pitch, knee, ankle
109           12, 13, 14, 15,   # R leg: hip_roll, hip_pitch, knee, ankle
110           16,               # torso
111           17, 20,           # L arm: shoulder_pitch, elbow
112           46, 49,           # R arm: shoulder_pitch, elbow
113       ]
114       joint_vel = data.qvel[:, qvel_idx]  # (B,13)
115
116       obs = torch.cat(
117           [
118               pelvis_z,            # (B,1)
119               gravity_in_base,     # (B,3)
120               heading_xy,          # (B,2)
121               v_body,              # (B,3)
122               w_body,              # (B,3)
123               speed_target,        # (B,1)
124               speed_error,         # (B,1)
125               foot_contact,        # (B,2)
126               lf_in_base,          # (B,3)
127               rf_in_base,          # (B,3)
128               joint_pos,           # (B,13)
129               joint_vel,           # (B,13)
130           ],
131           dim=-1,
```

```
132          )
133
134          return obs
```

**LOAM-Designed Action Model for `h1hand-run-v0`**

```
1    # Action expansion for h1hand-run-v0 (final)
2    # - Compact action_dim = 13 focused on locomotion and balance:
3    #   Legs (8): per leg [hip_roll, hip_pitch, knee, ankle] - hip_yaw fixed to 0
          for straight running.
4    #   Torso (1): torso actuator for balance.
5    #   Arms (4): per arm [shoulder_pitch, elbow] for rhythmic swing;
          shoulder_roll via mild synergy; shoulder_yaw/wrist_yaw fixed to 0.
6    # - Hands (20 actuators per both hands) are set to 0 (neutral) - not needed
          for running.
7    # - Mild synergy:
8    #     left_shoulder_roll = -0.2 * left_hip_roll_action
9    #     right_shoulder_roll = -0.2 * right_hip_roll_action
10   # - Batch/device safe with shape validation.
11   def compute_action(action, data):
12       import torch
13
14       # Normalize action input shape to (B, 13)
15       single = action.dim() == 1
16       if single:
17           action = action.unsqueeze(0)
18
19       B = action.shape[0]
20       action_dim = 13
21       if action.shape[-1] != action_dim:
22           raise ValueError(f"Expected action_dim {action_dim}, got {tuple(
          action.shape)}")
23
24       device = action.device
25       dtype = action.dtype
26
27       # Prepare output (B, 61)
28       action_out = torch.zeros((B, 61), device=device, dtype=dtype)
29
30       # Split low-dim actions
31       # Legs
32       l_leg = action[:, 0:4]   # [hip_roll, hip_pitch, knee, ankle]
33       r_leg = action[:, 4:8]   # [hip_roll, hip_pitch, knee, ankle]
34       # Torso
35       torso = action[:, 8:9]   # [torso]
36       # Arms
37       l_arm = action[:, 9:11]  # [shoulder_pitch, elbow]
38       r_arm = action[:, 11:13] # [shoulder_pitch, elbow]
39
40       # Mild balance synergy for shoulder roll from hip roll
41       l_sh_roll = -0.2 * l_leg[:, 0:1]
42       r_sh_roll = -0.2 * r_leg[:, 0:1]
43
44       # Map to 61 actuators
45       # Legs (0-9)
46       action_out[:, 0] = 0.0           # left_hip_yaw
47       action_out[:, 1] = l_leg[:, 0]   # left_hip_roll
48       action_out[:, 2] = l_leg[:, 1]   # left_hip_pitch
49       action_out[:, 3] = l_leg[:, 2]   # left_knee
```

```
50        action_out[:, 4] = l_leg[:, 3]      # left_ankle
51        action_out[:, 5] = 0.0              # right_hip_yaw
52        action_out[:, 6] = r_leg[:, 0]      # right_hip_roll
53        action_out[:, 7] = r_leg[:, 1]      # right_hip_pitch
54        action_out[:, 8] = r_leg[:, 2]      # right_knee
55        action_out[:, 9] = r_leg[:, 3]      # right_ankle
56
57        # Torso (10)
58        action_out[:, 10] = torso[:, 0]     # torso
59
60        # Arms (11-20)
61        action_out[:, 11] = l_arm[:, 0]     # left_shoulder_pitch
62        action_out[:, 12] = l_sh_roll[:, 0] # left_shoulder_roll (synergy)
63        action_out[:, 13] = 0.0             # left_shoulder_yaw
64        action_out[:, 14] = l_arm[:, 1]     # left_elbow
65        action_out[:, 15] = 0.0             # left_wrist_yaw
66        action_out[:, 16] = r_arm[:, 0]     # right_shoulder_pitch
67        action_out[:, 17] = r_sh_roll[:, 0] # right_shoulder_roll (synergy)
68        action_out[:, 18] = 0.0             # right_shoulder_yaw
69        action_out[:, 19] = r_arm[:, 1]     # right_elbow
70        action_out[:, 20] = 0.0             # right_wrist_yaw
71
72        # Hands (21-60) remain zeros (neutral/passive)
73
74        # Validate final shape
75        if action_out.shape != (B, 61):
76            raise RuntimeError(f"Expanded action has wrong shape {tuple(
              action_out.shape)}, expected {(B, 61)}")
77
78        # Restore original shape if single sample input
79        if single:
80            action_out = action_out.squeeze(0)
81
82        return action_out
```

### B.2.3  ANALYSIS OF LOAM-DESIGNED OBSERVATION MODEL (48 DIMENSIONS)

The LOAM-Designed model redesigns the observation space to be more compact and semantically meaningful. The function reduces the observation space from 151 to 48 dimensions by selecting only the joint states relevant to locomotion—legs, torso, and primary arm joints (lines 96-114). Proprioceptive states from the hands, which are passive during running, are excluded.

The observation is also designed to be invariant to the robot's heading, a modification that aids in learning generalizable locomotion policies. This is achieved by representing key vectors in the pelvis's local coordinate frame. For example, the world gravity vector (lines 66-67), base angular velocities (lines 77-78), and relative foot positions (lines 90-93) are all transformed into this local frame using quaternion rotations.

Furthermore, the function engineers features that directly relate to the task goal. It includes the target speed (5.0 m/s) and the current forward speed error as direct inputs to the policy (lines 68-69). This provides a low-latency signal correlated with the reward function. For stability, the design includes the positions of the feet relative to the pelvis (lines 88-93) and binary foot contact sensor data (line 85), which inform the policy about the robot's base of support and the current phase of the gait cycle.

### B.2.4  ANALYSIS OF LOAM-DESIGNED ACTION MODEL ($13 \rightarrow 61$ DIMENSIONS)

Complementing the observation space, the LLM structured the action space around a low-dimensional latent representation. The policy learns to control a 13-dimensional vector that is deterministically expanded to the full 61-dimensional actuator space. This latent vector corresponds to

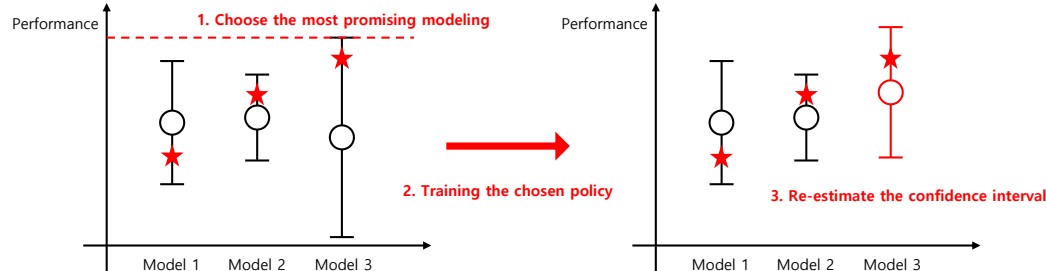

Figure 8: Illustration of the OFU-based selection process. At each iteration, (1) the model with the highest upper confidence bound is chosen, (2) its policy is further trained for a fixed step, and (3) the confidence interval is re-estimated. This optimistic rule ensures a balance between exploiting promising models and continuing to explore those that may turn out to be optimal.

the key joints for locomotion: 8 for the legs, 1 for the torso, and 4 for the primary arm swing (lines 30-38).

The design also embeds a biomechanical prior by hard-coding a contralateral coordination pattern. It couples the shoulder roll action to the opposite hip's roll action (lines 42-43), which is then mapped to the shoulder roll actuators (lines 41-42, 62 and 67). This design generates a stabilizing arm swing that counter-rotates the torso against leg movements, removing the need for the agent to learn this fundamental coordination from scratch. Finally, actuators for joints not directly involved in running, such as those in the hands or for hip/shoulder yaw, are fixed to zero (lines 28, 46, 51, 63, 65, 68, and 70). This focuses the agent's learning on the essential degrees of freedom for the gait.

## C    EXPLORATION-EXPLOITATION BASED ON OPTIMISM

To construct LOAM-Race, we adopt optimism-in-the-face-of-uncertainty (OFU), a widely used strategy for balancing exploration and exploitation. This OFU-based rule efficiently narrows the set of promising candidates while maintaining a natural trade-off between exploration and exploitation, as illustrated in Figure 8.

At each iteration, we estimate the upper confidence bound of every candidate model from its estimated performance and associated uncertainty, and then select the model with the highest bound. The chosen policy is then trained for a fixed step, which refines its performance estimate and reduces uncertainty. This selection rule inherently balances exploration—by occasionally sampling uncertain candidates—and exploitation—by favoring those with strong empirical performance.

## D    RAW OBSERVATION FEATURE

### D.1    HUMANOIDBENCH

While HumanoidBench provides the predefined observation features, qpos and qvel, we use 13 attributes in mj.Data, which are commonly employed in robot locomotion and manipulation simulations as well as RL environments, as follows:

- qpos: Generalized positions (joint angles, base pose).
- qvel: Generalized velocities (joint rates, base velocity).
- site_xpos: 3D position of each defined site (useful for end-effectors, sensors, etc.).
- site_xmat: 3x3 rotation matrix representing the orientation of each site (flattened).
- qfrc_actuactor: Actuator forces mapped to degrees of freedom.
- xpos: 3D position of each body.
- xquat: 4D orientation (quaternion) of each body.
- xmat: 3x3 rotation matrix representing the orientation of each body (flattened).

- `cvel`: 6D spatial velocity (linear & angular) of each body.
- `cfrc_ext`: 6D external wrench (force & torque) on each body.
- `cinert`: 10D composite rigid body inertia of each body.
- `actuator_force`: Force/torque generated by each actuator.
- `sensordata`: Scalar sensor outputs.

## D.2 ISAAC LAB

In our Isaac Lab experiments, which utilize the NVIDIA PhysX engine, the raw observation space ($O_{raw}$) provided to the LLM consists of physical states accessible through the `isaaclab.assets.RigidObjectData` interface.

- `joint_pos`: Joint positions of all joints.
- `joint_vel`: Joint velocities of all joints.
- `joint_acc`: Joint acceleration of all joints.
- `root_link_pos_w`: Root link position in world frame.
- `root_link_quat_w`: Root link orientation $(w, x, y, z)$ in world frame.
- `root_link_lin_vel_w`: Root link linear velocity in world frame.
- `root_link_ang_vel_w`: Root link angular velocity in world frame.
- `root_link_lin_vel_b`: Root link linear velocity in base frame.
- `root_link_ang_vel_b`: Root link angular velocity in base frame.
- `root_link_state_w`: Root link state [pos, quat, lin_vel, ang_vel] in world frame.
- `root_state_w`: Alias for root link state in world frame.
- `root_com_pos_w`: Root center of mass position in world frame.
- `root_com_quat_w`: Root center of mass orientation $(w, x, y, z)$ in world frame.
- `root_com_lin_vel_w`: Root center of mass linear velocity in world frame.
- `root_com_ang_vel_w`: Root center of mass angular velocity in world frame.
- `root_com_lin_vel_b`: Root center of mass linear velocity in base frame.
- `root_com_ang_vel_b`: Root center of mass angular velocity in base frame.
- `root_com_vel_w`: Root center of mass velocity [lin_vel, ang_vel] in world frame.
- `root_com_state_w`: Root center of mass state [pos, quat, lin_vel, ang_vel] in world frame.
- `body_link_pos_w`: Positions of all bodies' link frames in world frame.
- `body_link_quat_w`: Orientation $(w, x, y, z)$ of all bodies' link frames in world frame.
- `body_link_lin_vel_w`: Linear velocity of all bodies' link frames in world frame.
- `body_link_ang_vel_w`: Angular velocity of all bodies' link frames in world frame.
- `body_link_vel_w`: Body link velocity [lin_vel, ang_vel] in world frame.
- `body_link_state_w`: State of all bodies' link frame [pos, quat, lin_vel, ang_vel] in world frame.
- `body_state_w`: Alias for body link state in world frame.
- `body_com_pos_w`: Positions of all bodies' center of mass in world frame.
- `body_com_quat_w`: Orientation $(w, x, y, z)$ of the principle axis of inertia of all bodies in world frame.
- `body_com_lin_vel_w`: Linear velocity of all bodies' center of mass in world frame.
- `body_com_ang_vel_w`: Angular velocity of all bodies' center of mass in world frame.
- `body_com_vel_w`: Body center of mass velocity [lin_vel, ang_vel] in world frame.

- `body_com_state_w`: State of all bodies' center of mass [pos, quat, lin_vel, ang_vel] in world frame.
- `projected_gravity_b`: Projection of the gravity direction on the base frame.
- `heading_w`: Yaw heading of the base frame (in radians).
- `generated_commands`: The desired goal pose for the target object (e.g., cube) specified in the environment (world) frame.

# E  MORE DETAILS OF ISAAC LAB EXPERIMENTS

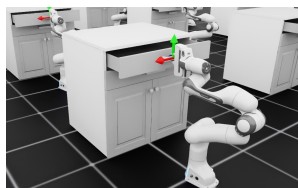 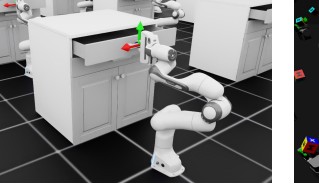 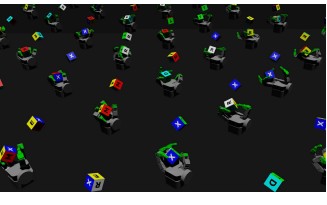

(a) Isaac-Lift-Cube-Franka-v0  (b) Isaac-Open-Drawer-Franka-v0  (c) Isaac-Repose-Cube-Allegro-v0

Figure 9: Examples of Isaac Lab tasks.

Beyond the MuJoCo simulation engine and Humanoid morphologies, we extend our evaluation to NVIDIA Isaac Lab (Mittal et al., 2023). This environment utilizes the PhysX simulation engine, offering a distinct simulation dynamics compared to MuJoCo. We test LOAM on three challenging tasks involving widely used robotic platforms: the Franka Emika Panda (7-DoF arm) and the Allegro Hand (16-DoF dexterous hand).

We evaluated LOAM on the following tasks (Figure 9):

- **Isaac-Lift-Cube-Franka-v0**: Pick a cube and bring it to a sampled target position with the Franka robot. (handcrafted observation: 36 dimension, raw observation: 1,152 dimension)
- **Isaac-Open-Drawer-Franka-v0**: Grasp the handle of a cabinet's drawer and open it with the Franka robot. (handcrafted observation: 31 dimension, raw observation: 1,801 dimension)
- **Isaac-Repose-Cube-Allegro-v0**: In-hand reorientation of a cube using Allegro hand. (handcrafted observation: 72 dimension, raw observation: 1,928 dimension)

In our Isaac Lab experiments, we define the raw observation space ($O_{raw}$) to consist solely of fundamental physical states accessible directly through the `isaaclab.assets.RigidObjectData` API as in HumanoidBench, while FastTD3 uses a handcrafted observation model provided by Isaac Lab. We include **every** dynamic state variable available in the API—excluding only static constants and mathematically identical duplicates—thereby exposing the LLM to the complete, uncurated physical state of the system. A comprehensive list of these features is provided in Appendix D.2.

Note that we intentionally exclude high-level, pre-processed features—typically provided in default Isaac Lab environments—from the raw observation set. For instance, the default (handcrafted) observation space for the Allegro Hand task includes computed values such as `goal_quat_diff`—a derived feature that explicitly calculates the quaternion difference between the current and target orientations to simplify the learning problem.

In contrast, LOAM is restricted to raw absolute data, such as the root link's orientation in the world frame (`root_link_quat_w`) and the target command (`generated_commands`). Consequently, the LOAM-generated observation function is required to autonomously discover and implement the necessary mathematical transformations (e.g., quaternion multiplication for relative orientation) to extract task-relevant information, rather than relying on human-engineered shortcuts.

## F    ABLATION STUDIES

### F.1    COMPARISON IN LLM BACKBONE

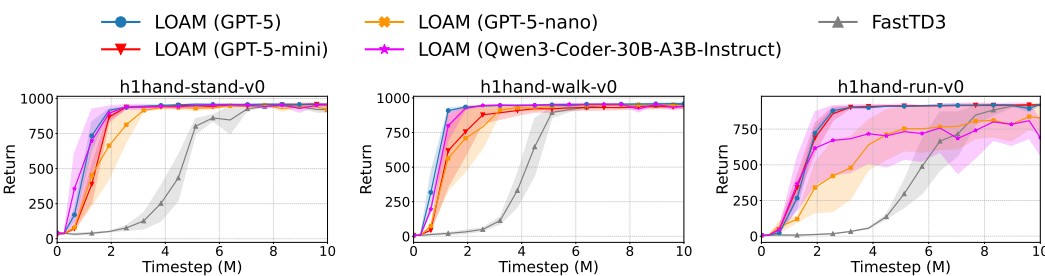

Figure 10: Learning curves of LOAM with different LLM backbones on HumanoidBench `stand`, `walk`, and `run` tasks.

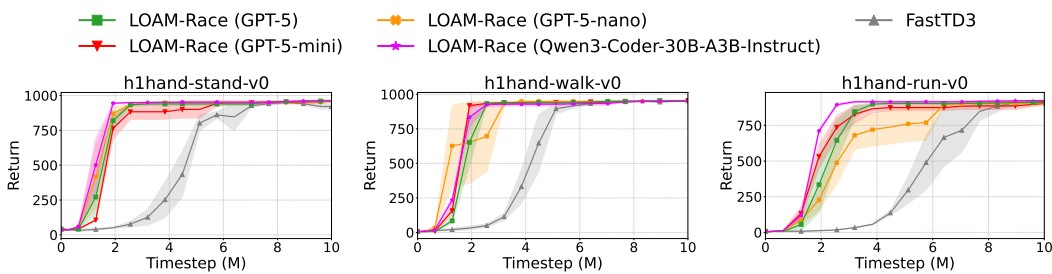

Figure 11: Learning curves of LOAM-Race with different LLM backbones on HumanoidBench `stand`, `walk`, and `run` tasks.

To address concerns regarding the dependency of our framework on state-of-the-art proprietary models (e.g., GPT-5), we conducted an ablation study using varying LLM sizes and an open-weights model. We compared the standard GPT-5 [5] against lighter variants (GPT-5-mini [6], GPT-5-nano [7]) and the open-weights Qwen3-Coder-30B-A3B-Instruct (Yang et al., 2025), analyzing their performance relative to the FastTD3 baseline.

As illustrated in Figures 10 and 11, a consistent pattern emerges across all tasks: regardless of the LLM backend employed—from the large-scale GPT-5 to the compact GPT-5-nano—both LOAM and LOAM-Race substantially outperform the non-LLM baseline (FastTD3). This demonstrates that semantic guidance from LLMs, even those with constrained capacity, provides a fundamental advantage in accelerating policy learning compared to approaches relying exclusively on handcrafted observation-action models.

However, while performance remains robust on simpler tasks (e.g., `stand` and `walk`), we observe a notable divergence in the more challenging `run` task under the standard LOAM framework (Figure 10). Specifically, models with reduced reasoning capabilities—such as GPT-5-nano and Qwen3-Coder-30B—exhibit marked performance degradation and delayed convergence compared to GPT-5. The gap widens significantly after approximately 4M timesteps, with GPT-5-nano and Qwen3-Coder plateauing at lower asymptotic returns.

---

[5]The flagship model with the highest parameter count and reasoning capability. We use this as the upper-bound baseline to evaluate the maximum potential of our framework.

[6]A mid-sized model optimized for a balance between cost and performance. It retains significant reasoning abilities while offering lower latency compared to the GPT-5.

[7]The most lightweight variant, designed for extreme efficiency and low-latency applications. While it has the lowest reasoning capacity among the three, it serves to test the feasibility of our method in resource-constrained or on-device settings.

Critically, the LOAM-Race framework (Figure 11) not only addresses this sensitivity to LLM capability but, remarkably, enables smaller models to match or even exceed the performance of their larger counterparts. In the `run` task, LOAM-Race successfully narrows the performance gap, with the open-weights Qwen3-Coder-30B achieving returns comparable to—and in some phases surpassing—GPT-5. More strikingly, in the `stand` task, Qwen3-Coder-30B consistently outperforms all proprietary models throughout training, demonstrating that competitive racing dynamics can effectively leverage the capabilities of open-source LLMs. These results underscore a key contribution of the LOAM-Race mechanism: it enhances framework robustness by reducing dependence on high-capacity LLMs while simultaneously unlocking the full potential of smaller models through collaborative refinement, thereby demonstrating the practical viability of deploying our method with more accessible, cost-effective, and open-source language models.

## F.2 COMPARISON IN ACQUISITION FUNCTIONS OF LOAM-RACE

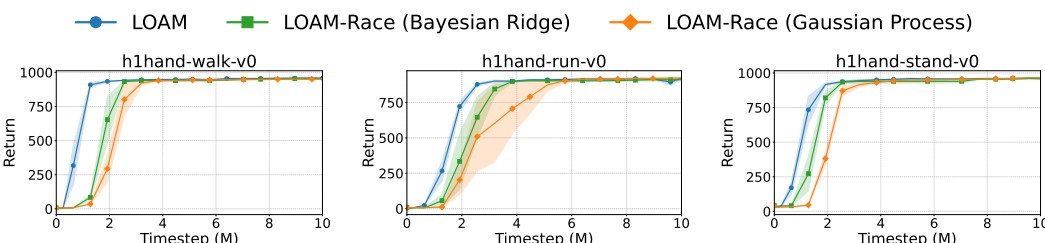

Figure 12: Learning curves of LOAM with different acquisition functions on HumanoidBench `stand`, `walk`, and `run` tasks.

In the main experiment, LOAM-Race utilizes **Bayesian Ridge Regression (BRR)** as an acquisition function to select a candidate model. This choice was motivated by its computational efficiency and robustness to noise in linear trends. To verify if a more complex estimator yields better selection performance, we replaced the BRR with a **Gaussian Process Regressor (GPR)** using an RBF kernel, which can capture non-linear trajectory patterns. We utilized Scikit-learn (Pedregosa et al., 2011) to implement both acquisition functions.

We evaluated both functions on the basic locomotion tasks (`stand`, `walk`, `run`) in HumanoidBench. As illustrated in Figure 12, the results indicate that the choice of the function does not significantly impact the final performance. Both BRR and GPR converge to nearly identical asymptotic returns across all tasks. In *run* task, the BRR actually demonstrates slightly faster convergence and lower variance in the early training phases compared to the GPR.

## F.3 RACING VS. ITERATIVE REFINEMENT

To validate the design choice of our candidate selection mechanism, we compare our proposed method, **LOAM-Race**, against an alternative iterative strategy named **LOAM-Refine**. Unlike LOAM-Race, which evaluates candidates in parallel, LOAM-Refine attempts to iteratively improve the observation and action space definitions during training.

A refinement step is triggered if the agent's return fails to exceed the current maximum for 5 consecutive checkpoints. Modifying the observation or action specifications inevitably alters the input or output dimensions (or semantics) of the underlying policy and value networks. Since transferring learned weights between neural networks with mismatched architectures is non-trivial, we re-initialize the agent's parameters and restart training from scratch after each refinement.

The experimental results in Figure 13 highlight the inherent limitations of this iterative approach. As shown in Seed 0 (Figure 13a) and Seed 2 (Figure 13c), the mandatory resets cause a fragmentation of the total training budget. While LOAM-Race allows a candidate to utilize the full 50M timesteps to reach asymptotic convergence, LOAM-Refine splits this budget into shorter segments. The agent is repeatedly interrupted just as it begins to learn, losing all prior experience and being forced to relearn basic locomotion skills from scratch. This inefficiency prevents the policy from mastering complex behaviors within the limited time remaining after a reset.

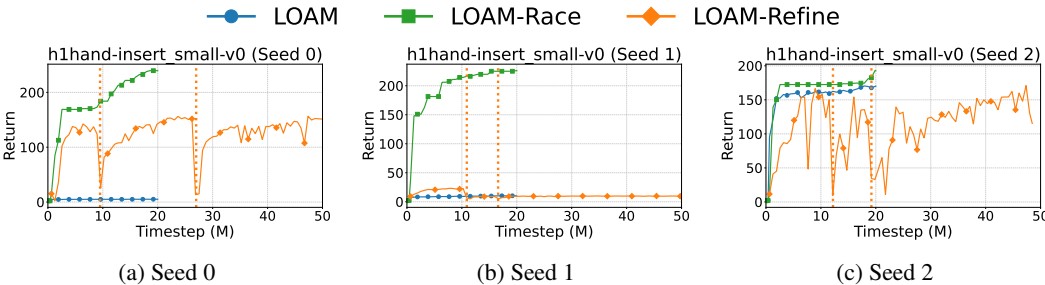

(a) Seed 0          (b) Seed 1          (c) Seed 2

Figure 13: Performance comparison between LOAM-Race (Ours) and LOAM-Refine. The green line represents the average performance of LOAM-Race, while the orange line shows individual runs of LOAM-Refine across three random seeds. Orange vertical dotted lines indicate the timesteps where the observation-action space was refined.

Furthermore, the iterative strategy proves ineffective when the initial candidate is fundamentally flawed. In Seed 1 (Figure 13b), where the starting design yields near-zero returns, subsequent refinements fail to recover a viable solution. The combination of a poor starting point and the sample inefficiency of repeated resets results in a complete failure, confirming that iteratively searching for a design is less robust than LOAM-Race's parallel selection. By generating candidates upfront and evaluating them without structural interruptions, LOAM-Race ensures that the best candidate benefits from the maximum available sample budget, avoiding the wasteful cold starts associated with iterative refinement.

### F.4 ADDITIONAL GUIDANCE IN PROMPT

We append additional guidance to prompts used for LLM-based observation and action modeling. First, we modify the task descriptions in HumanoidBench into more precise definitions. For example, in 'h1hand-reach-v0', the description *"Reach a randomly initialized 3D point with the left hand."* can be misread as involving only the hand, whereas the task requires whole-body movement toward a distant target. Without clarification, an LLM may ignore the rest of the body in observation and action modeling. To prevent this, we restate the task as ***"Touch a target point in space using its left hand. It must accomplish this while maintaining a stable and upright posture."*** Second, we add a line to reasoning guideline, emphasizing that whole-body posture is an implicit requirement for any physical task; ***"For any physical task, maintaining the robot's stable and balanced posture is a fundamental and implicit requirement. The final code must reflect a deep understanding of the need for whole-body postural control while performing the primary task."***. We observed that these changes ensure the LLM interprets and models the task correctly, capturing both the target objective and the necessary full-body constraints, resulting in higher performance across some tasks such as reach (Figure 14).

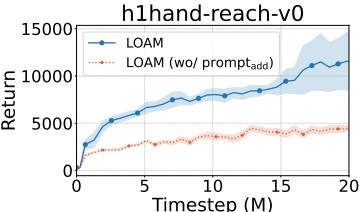

Figure 14: Ablation on the additional guidance in prompts.

