# OpenReview forum: "Designing Observation and Action Models for Efficient Reinforcement Learning with LLMs"
_ICLR.cc/2026/Conference — Submitted to ICLR 2026_

### Official Review · Reviewer_A8cU · 2025-10-30

**Soundness:** 2
**Presentation:** 2
**Contribution:** 2
**Rating:** 4
**Confidence:** 4

**Summary:**

The paper proposes LOAM, which uses an LLM to automatically generate the observation and action functions that define how an RL agent perceives the environment and issues commands. Instead of relying on handcrafted mappings, LOAM creates these interfaces in code and plugs them into existing training pipelines. A second component, LOAM-Race, samples multiple candidate designs, briefly trains each one, and then reallocates the fixed training budget to the most promising candidates. Experiments across 12 HumanoidBench tasks show faster convergence and, in some cases, better performance than other baselines.

**Strengths:**

1.1: Addresses an underexplored component in RL, which is the design of observation and action mappings.

1.2: Strong experimental results with different environments (same domain) with multiple seeds.

**Weaknesses:**

2.1: No optimization in the design space. The method goes from zero-shot to train to pick. There is no iterative refinement or evolution of the generated code (obs/action functions).

2.2: LOAM-Race is not clearly described. The figure mentions selection “every ~128k steps,” but the paper does not specify the details. The results show that LOAM-Race has the same total training steps as the other baselines, so it is unclear how the total training budget is divided.

2.3: The discard rule of weakest every 128k steps can be wrong. More complicated functions require more training time to converge and may lead to better results. That should be investigated.

2.4: The claim that LOAM-Race “handles LLM stochasticity” is misleading. The approach samples multiple candidates and reallocates training based on performance every 128K steps.

2.5: Limited domain diversity. All experiments are on humanoid tasks; the method should be tested on other domains or harder environments to verify generality and performance beyond convergence speed.

2.6: Section-3 is overly detailed with prompt templates that belong in the appendix. The space should instead expand on 2.2 (racing/budget details).

**Questions:**

3.1: Can the authors provide more information about the overall pipeline design and justify why there is no optimization or refinement over the generated code space (i.e., beyond zero-shot generation and simple selection among K candidates)?

3.2: Can you show additional results or analysis verifying whether weaker early-performance candidates in LOAM-Race eventually lead to worse final policies, or if early performance reliably predicts long-term outcomes?

3.3: Can you add additional results in more domains where LOAM (and/or LOAM-Race) outperforms other baselines consistently, not only in speed but in performance as well?

---

> ### Author Response · Authors · 2025-11-26
>
> Thank you for your valuable and constructive feedback.
> We have revised the explanation of LOAM-Race in Section 4.2 of the updated manuscript. For a detailed description, please refer to Section 4.2 of the revised paper.
>
> **Response to W1, Q1: Justification for Zero-shot Generation vs. Iterative Refinement**
>
> While iterative refinement is possible in principle, our empirical experience shows that it is rarely effective: if the initial model fails, subsequent refinements tend to fail as well, whereas generating a new model from scratch is typically more reliable.
> We include an ablation study on refinement in the revised paper. For further details, please refer to Appendix F.3.
>
> **Response to W2: Clarification on LOAM-Race Mechanism**
>
> For LOAM-Race, each candidate first receives the same race timesteps (128K steps in our experiments). After that, LOAM-Race repeatedly allocates a race timestep to the most promising candidate. At the end of each iteration, the acquisition scores are recomputed and the next promising model is selected. This process continues until the maximum timestep (i.e., the standard single-model training timesteps) is fully consumed.
> To clarify this procedure, we provide a more detailed explanation of the LOAM-Race mechanism including pseudocode for LOAM-Race (Algorithm 1) in the revised manuscript.
>
> **Response to W3, Q2: Concern with 'discard the weakest'**
>
> LOAM-Race does not enforce a 'discard the weakest' rule. Unlike approaches that permanently eliminate low-performing models (hard pruning), we maintain all candidates throughout the entire process.
> Our acquisition function explicitly balances exploitation (expected performance) and exploration (predictive uncertainty). As a leading candidate undergoes further training, its uncertainty term naturally diminishes. This reduction allows less-sampled candidates—whose uncertainty remains high—to regain competitiveness, ensuring that potentially strong models are not prematurely discarded due to early stochasticity.
> For instance, in Figure 5 (c), Env Model 2 also receives essential allocation to identify its potential.
>
> **Response to W4: Clarification on Handling Stochasticity**
>
> We note that LOAM-Race does not eliminate the inherent stochasticity of LLM outputs. Instead, it alleviates its impact by evaluating multiple candidates. We have revised the manuscript to clarify this point.
>
> **Response to W5, Q3: Generalization to New Domains**
>
> We have additionally evaluated LOAM on tasks in NVIDIA Isaac Lab, which provides a different environment from the HumanoidBench tasks used in the main experiments.
> In this environment, we assess on three representative tasks: Lift-Cube, Open-Drawer, and Repose-Cube. Across these tasks, both LOAM and LOAM-Race achieve promising performance, indicating that LOAM is not restricted to HumanoidBench and can transfer to other domains beyond our primary setting. For details, please refer to Sections 6.3 of the revised paper, where we include the corresponding setup and experimental results.
>
> **Response to W6: Structural Revisions**
>
> We have revised Section 3 accordingly and expanded the explanation of LOAM-Race in Section 4.2 of the updated manuscript.

---

### Official Review · Reviewer_4CzN · 2025-11-01

**Soundness:** 3
**Presentation:** 3
**Contribution:** 2
**Rating:** 6
**Confidence:** 3

**Summary:**

Disclosure: Claude is used to refine this review.

This paper introduces LOAM (LLM-based design of Observation and Action Models), a framework that leverages large language models to automatically generate observation and action representations for reinforcement learning tasks. Given environment specifications and task descriptions, LOAM produces Python functions that transform raw sensory inputs into compact observation vectors and map low-dimensional policy outputs to full actuator commands. The authors also propose LOAM-Race, which handles LLM output variability by racing multiple generated designs and progressively selecting top performers. Experiments on HumanoidBench demonstrate that LOAM-designed models achieve approximately 3× faster learning on basic locomotion tasks compared to handcrafted baseline models.

**Strengths:**

- The paper addresses an important yet understudied problem in reinforcement learning - automated design of observation and action spaces, which practitioners typically handle through manual feature engineering. The core insight that LLMs can automate this process is compelling and timely.
- The empirical results are strong, with consistent improvements across multiple tasks in HumanoidBench, and the reach task result is particularly impressive where LOAM succeeds while all baselines fail.
- The paper provides extensive implementation details, including full prompt templates and generated code examples in the appendices, which aids reproducibility and understanding.

**Weaknesses:**

- The evaluation scope is limited to a single benchmark (HumanoidBench), so it's unclear if it can generalize to different robots or task domains.
- The prompts appear heavily engineered with domain-specific guidance, which contradicts claims of automation and suggests significant manual tuning was required. Maybe a reasonable comparison is how much human effort it saves (i.e., how many human hours are needed to match the performance of the proposed method).

**Questions:**

N/A

---

> ### Author Response · Authors · 2025-11-26
>
> Thank you for your insightful and constructive feedback.
>
> **Response to W1: Validation of Generalizability beyond HumanoidBench**
>
> We have additionally evaluated LOAM on tasks in NVIDIA Isaac Lab, which provides a different environment from the HumanoidBench tasks used in the main experiments.
> In this environment, we assess on three representative tasks: Lift-Cube, Open-Drawer, and Repose-Cube. Across these tasks, both LOAM and LOAM-Race achieve promising performance, indicating that LOAM is not restricted to HumanoidBench and can transfer to other domains beyond our primary setting. For details, please refer to Sections 6.3 of the revised paper, where we include the corresponding setup and experimental results.
>
> **Response to W2: Concerns on Heavily Engineered Prompts**
>
> We would like to emphasize that we do not provide heavily-engineered domain-specific guidance for LLM. For Isaac Lab experiments, we provide only the simple task descriptions available in Isaac Lab Documents [1], which are comparable in level of detail to the brief descriptions used for HumanoidBench. We also define the raw observation space ($O_{raw}$) to consist solely of fundamental physical states that are directly accessible through the `isaaclab.assets.RigidObjectData API` (Appendix D.2 of the revised paper), following the convention used in HumanoidBench. In addition, most prompts are shared across all tasks, while task-specific information, such as the task description, should be provided as appropriate for each task.
>
> [1] Task description: https://isaac-sim.github.io/IsaacLab/main/source/overview/environments.html

---

### Official Review · Reviewer_snyz · 2025-11-01

**Soundness:** 2
**Presentation:** 3
**Contribution:** 3
**Rating:** 4
**Confidence:** 3

**Summary:**

LOAM uses LLMs to automatically generate Python functions for observation and action models in RL, based on environment specs and tasks, enabling efficient integration into training pipelines. LOAM-Race races multiple variants to select the best under fixed budgets. Evaluated on HumanoidBench (12 locomotion/manipulation tasks) with FastTD3, achieving 3x faster learning and higher returns than baselines like handcrafted features and LESR.

**Strengths:**

1.Automates a key RL bottleneck (obs/action design) with LLMs, yielding compact, task-relevant models that boost sample efficiency and final performance across diverse tasks.
2.LOAM-Race effectively mitigates LLM output variability via optimistic racing, identifying strong designs in a single run without exhaustive training.
3.Structured prompts incorporate robotics priors (e.g., posture stability), enhancing model quality as shown in ablations.

**Weaknesses:**

1.No quantitative LLM cost analysis—racing requires multiple generations/evaluations, potentially prohibitive for larger setups.
2.Baselines (e.g., LESR) are adapted but may not be optimally tuned; lacks comparisons to non-LLM obs/action methods.
3.Results confined to simulation; real-robot gaps (noise, delays) unaddressed, limiting claims of practical utility.
4.Over-reliance on proprietary GPT-5 without testing open-source alternatives or model robustness.

**Questions:**

1.How does LOAM perform with open-source LLMs (e.g., Llama-3) versus GPT-5? Any degradation in model quality?
2.What are the total LLM inference costs (tokens, time) for generating and racing models per task?
3.Can LOAM handle visual or partial observations, e.g., by incorporating neural encoders?
4.Why no ablation on racing hyperparameters like K (candidates) beyond K=1-4, or confidence estimation methods?
5.Have you tested LOAM on non-MuJoCo envs or real hardware to validate beyond simulation?

---

> ### Author Response · Authors · 2025-11-26
>
> Thank you for your valuable and constructive feedback.
>
> **Response to W1, Q2: Quantitative Analysis of LLM Inference Costs and Time Efficiency**
>
> We measure the costs across three representative tasks: `Stand` (Locomotion), `Cube` (Static Manipulation), and `Powerlift` (Dynamic Manipulation), each selected as a canonical example from the three task categories defined in our main experiments. Averaged over these three tasks, generating a complete environment design (observation + action) costs approximately $0.25 and takes about 6.7 minutes.
>
> For LOAM-Race, which generates and evaluates 3 candidates in parallel, the total cost is simply $3\times$ that of a single design, resulting in approximately $0.74 and 20 minutes per task. These costs are negligible compared to the total wall-clock training time of continuous-control RL (typically hours). Importantly, no additional LLM calls are made during training or racing procedure, so the overall LLM cost remains small and does not meaningfully impact scalability.
>
> **Response to W2: Fairness of Baseline Comparisons**
>
> We would like to clarify that we made efforts to ensure a fair comparison with LESR, particularly by (i) sharing the same RL backbone, and (ii) allocating additional pilot-evaluation budget.
>
> First, the key difference between LESR and LOAM(-Race) lies in how they use LLMs to design the model, rather than in the underlying RL training procedure. To ensure a fair evaluation, we therefore replace LESR’s original TD3 implementation with FastTD3, which is known for its sample efficiency in robotic tasks.
>
> Finally, LESR requires pilot evaluations to select the best candidate, which introduces additional environment steps. We found that each pilot evaluation requires approximately $0.5\times T$, where $T$ is the total number of training timesteps for a standard single-model training, for stable selection. As LESR evaluates nine candidates (three initial candidates $\times$ three refinement rounds), this results in $4.5\times T$ for pilot selection. After, LESR performs another $T$ steps for final policy training. Overall, LESR uses approximately $5.5\times T$ steps while LOAM(-Race) uses $T$ steps.
>
> **Response to W3, Q5: Generalization to Non-MuJoCo Environments and Sim-to-Real.**
>
> We have additionally evaluated LOAM on tasks in NVIDIA Isaac Lab, which provides a different environment from the HumanoidBench tasks used in the main experiments.
> In this environment, we assess on three representative tasks: Lift-Cube, Open-Drawer, and Repose-Cube. Across these tasks, both LOAM and LOAM-Race achieve promising performance, indicating that LOAM is not restricted to HumanoidBench and can transfer to other domains beyond our primary setting. For details, please refer to Sections 6.3 of the revised paper, where we include the corresponding setup and experimental results.
>
> Note that LOAM is a framework for general RL. Once an observation-action model (or a reward model, in related work) is produced, an RL algorithm is trained on top of it. This stage occurs prior to deployment. Accordingly, our focus is aligned with prior work on generation of environment models for improving sample efficiency, rather than with sim-to-real research. Sim-to-real studies address challenges that arise when transferring a policy trained in simulation to the real world.
>
> **Response to W4, Q1: Robustness Across Different LLM Backbones**
>
> We additionally evaluate LOAM with GPT-5-mini, GPT-5-nano, and Qwen3-Coder-30B, and observe performance comparable to that achieved with GPT-5. This demonstrates that LOAM is not strongly dependent on a specific LLM. See the Section 6.4 of the revised paper, where we report the corresponding ablation results.

---

> ### Author Response · Authors · 2025-11-26
>
> **Response to Q3: Applicability to Partial Observability and Visual Inputs**
>
> LOAM already operates on partial, raw observations provided by the environment. In our setting, partial observability is the default, and LOAM’s generated observation models are designed to select, transform, and reshape these partial inputs, rather than assuming access to full state information.
>
> On the other hand, vision-based tasks are outside the scope of our work, as feature extraction from images is already handled effectively by existing CNN/ViT-based encoders. Nonetheless, integrating LOAM with standard visual encoders could be a promising direction for future research.
>
> **Response to Q4: Ablation Studies**
>
> We also conduct an additional ablation study on acquisition function (i.e., confidence estimation methods in review) using Gaussian process regression. In this experiment, both Bayesian ridge regression and Gaussian process regression yield nearly identical asymptotic returns. For details, please refer to Section 6.4 of the revised paper, where we report the corresponding ablation results.
>
> In our experiments, performance is typically highest at $K=3$ and degrades when $K\ge4$, as shown in Figure 5 (b). This is consistent with the fact that LOAM-Race operates under the same total number of training timesteps as LOAM.
> Large $K$ enforces to spend lots of timesteps to identify the potential of each candidate, which leads to a lack of the budget to train the policy for the best model.

---

### Official Review · Reviewer_fque · 2025-11-07

**Soundness:** 2
**Presentation:** 2
**Contribution:** 2
**Rating:** 4
**Confidence:** 4

**Summary:**

This paper introduces LOAM (LLM-based design of Observation and Action Models), a novel framework that leverages Large Language Models (LLMs) to "automate the generation" of observation and action models for reinforcement learning agents, particularly in complex domains like humanoid robotics. The core idea is to use structured prompts—containing information about the agent's morphology , the "task description" , and "available state variables" —to guide an LLM to generate Python functions (compute_obs and compute_action). These functions serve as a "wrapper" around the original environment , creating a lower-dimensional and more informative state-action space for the RL agent. To address the "inherent stochasticity" of LLM outputs, the authors also propose LOAM-Race, a mechanism based on the "principle of optimism in the face of uncertainty (OFU)" that efficiently evaluates multiple generated models in parallel and allocates training resources to the most promising candidates. The method is evaluated on the HumanoidBench benchmark, where it demonstrates significant improvements in sample efficiency and final performance over strong baselines.

The paper's claimed contributions are exceptional. The proposed method demonstrates strong empirical performance on the challenging HumanoidBench benchmark. This suggests a true qualitative leap in capability, not merely a quantitative improvement. The qualitative analysis in Appendix B further reveals that the LLM (allegedly) generates sophisticated, domain-aware Python code that embeds complex physical priors, such as "heading-invariance" via quaternion rotations and "biomechanical priors" like contralateral coordination.

**Strengths:**

1. The paper tackles a well-known and significant bottleneck in applying RL to complex robotic systems: the design of observation and action spaces. The proposed approach of using LLMs to automate this process is novel, timely, and presents a compelling new direction for environment design in RL.
2. The experimental results, as presented, are impressive. Achieving "over 3x faster learning on average" on a challenging benchmark like HumanoidBench (Figure 1) is a substantial improvement. The learning curves in Figure 4 clearly demonstrate that LOAM and LOAM-Race consistently and significantly outperform strong baselines like FastTD3 and LESR across a wide variety of tasks. The qualitative win on the h1hand-reach-v0 task is particularly noteworthy, suggesting the framework can discover representations superior to human-engineered ones.
3. The LOAM-Race mechanism is an intelligent and practical solution to the inherent stochasticity of LLM code generation. Instead of viewing variability as a weakness, the authors turn it into an opportunity for robust model selection. The method, based on the principle of optimism in the face of uncertainty, is principled and shown to be effective at finding better and more stable solutions.

**Weaknesses:**

1. The work primarily automates the implementation of the wrapper, not the conceptual design of the task. The LLM is effectively "automating the translation of a detailed human specification into code." The framework's success hinges on access to the pre-existing, human-engineered structure of the HumanoidBench environment and a "well-defined reward signal", a limitation the authors admit in the conclusion.  The LLM does not operate from raw physics but from a curated set of variables and, most importantly, a human-scripted reward function for each task. The LLM is given the task description and reward logic, which drastically simplifies the problem of identifying relevant features. The paper frames this as "automating design," but it feels more like "automating the translation of a detailed human specification into code." The framework's success hinges on access to a "well-defined reward signal" , a limitation the authors admit in the conclusion. This raises significant questions about its utility in more realistic scenarios where the reward is sparse or the task goal is not so clearly defined.
2. While simulation is a necessary first step, the paper makes strong claims about solving a key bottleneck for real-world robotics without providing any experiments or even a substantive discussion on the challenges of transferring these generated models to physical hardware. LLM-generated code might create brittle policies that overfit to simulation-specific dynamics or artifacts. An analysis of the sim-to-real gap would be essential for a paper with such practical claims.

**Questions:**

See Weakness.

---

> ### Author Response · Authors · 2025-11-26
>
> Thank you for your thoughtful and constructive feedback.
>
> **Response to W1**
>
> We would like to emphasize that we do not provide curated dataset for LLM. For Isaac Lab experiments, we provide only the simple task descriptions available in Isaac Lab Documents [1], which are comparable in level of detail to the brief descriptions used for HumanoidBench. We also define the raw observation space ($O_{raw}$) to consist solely of fundamental physical states that are directly accessible through the `isaaclab.assets.RigidObjectData API` (Appendix D.2 of the revised paper), following the convention used in HumanoidBench. In addition, most prompts are shared across all tasks, while task-specific information, such as the task description, should be provided as appropriate for each task.
>
> To clarify the scope of our work, LOAM generates the observation and action models, thereby replacing a component that has traditionally been an expertise-intensive bottleneck in RL with LLM generations.
>
> To avoid misunderstanding, we kindly note that LOAM does not have access to a well-defined human engineered reward function during generating observation and action models. LOAM uses only simple task descriptions and available variable information in the environment, rather than any curated variables or human-engineered reward structure. To address this misunderstanding, we have revised the overall manuscript to clarify the scope of LOAM.
>
> **Response to W2: Feasibility of Real-World Deployment**
>
> Note that LOAM is a framework for general RL. Once an observation-action model (or a reward model, in related work) is produced, an RL algorithm is trained on top of it. This stage occurs prior to deployment. Accordingly, our focus is aligned with prior work on generation of environment models for improving sample efficiency, rather than with sim-to-real research. Sim-to-real studies address challenges that arise when transferring a policy trained in simulation to the real world.
>
> [1] Task description: https://isaac-sim.github.io/IsaacLab/main/source/overview/environments.html

---

### Official Review · Reviewer_17hf · 2025-11-12

**Soundness:** 3
**Presentation:** 4
**Contribution:** 3
**Rating:** 6
**Confidence:** 4

**Summary:**

The paper introduces LOAM (LLM-based design of Observation and Action Models), a framework that automates the design of observation and action representations in reinforcement learning (RL). Instead of relying on handcrafted feature and actuator mappings, LOAM uses large language models (LLMs) to generate executable Python functions that define observation and action models. It also introduces LOAM-Race a model selection mechanism that evaluates multiple LLM-generated candidates in parallel and adaptively allocates training resources to the most promising ones using upper-confidence-bound (UCB) estimates.

Applied to **HumanoidBench**, a high-dimensional humanoid control benchmark, LOAM achieves up to **3× faster learning** than expert-designed models using the same RL algorithm (**FastTD3**). LOAM-Race further improves robustness by mitigating the variability of LLM-generated designs.

**Strengths:**

- **Novel contribution:** Automates a fundamental but underexplored RL design component, observation and action model specification using LLMs.
- **Clear methodology:** The structured prompting framework for code generation (system, observation, and action prompts) is systematic and well-motivated.
- **Strong empirical results:** Demonstrates consistent gains across locomotion and manipulation tasks on HumanoidBench, surpassing FastTD3 and LESR baselines.
- **Comprehensive experiments:** Includes detailed ablations on observation-only vs. full design, candidate count, and racing behavior, alongside qualitative code analysis.
- **Reproducibility:** Provides complete templates, prompts, and structured pipeline descriptions, making the approach easily replicable.

**Weaknesses:**

- **Limited theoretical grounding:** The paper is largely empirical and it lacks formal analysis of why LLM-generated designs improve representation quality or exploration.
- **Dependence on LLM reliability:** Quality and efficiency depend on the correctness of generated code while the robustness under different model types (e.g., GPT-4 vs. GPT-5) is not explored.
- **Limited scope of environments:** Focuses solely on humanoid control in simulation. Additional results on other domains (e.g., vision-based or multi-agent tasks) would test generality.
- **Novelty overlap:** Shares conceptual territory with recent LLM-for-environment design papers such as **LESR (Wang et al., 2024)**, **ExploRLLM (Ma et al., 2024)**, **Eureka (Ma et al., 2023)**, and **Text2Reward (Xie et al., 2023)**. The distinction lies mainly in targeting observation and action models rather than rewards.
- **No real-world validation:** Physical robot experiments or noisy sensory settings would significantly strengthen claims of practical impact.


LOAM-Race is claimed to be the first use of LLM and checking different candidates. ExploRLLM (https://arxiv.org/abs/2403.09583) also does that. They also shape observation and action spaces.

**Questions:**

Along with weaknesses:
1. How does LOAM generalize to non-robotic domains (e.g., grid-worlds or visual navigation)?
2. How sensitive is LOAM to LLM type and prompting format? Would smaller models (e.g., GPT-3.5) produce usable models?
3. Could LOAM-Race be extended to also handle reward model design simultaneously?
4. How does LOAM ensure the generated code’s physical plausibility (e.g., avoiding unfeasible joint mappings)?

---

> ### Author Response · Authors · 2025-11-26
>
> Thank you for your valuable and constructive feedback.
>
> **Response to W1, Q4: Theoretical Grounding and Physical Plausibility**
>
> While a full theoretical analysis is beyond this work's scope, we note that prior studies demonstrate LLMs can generate robot policy code with non-trivial geometric reasoning [1] and construct multibody dynamics models from natural language [2]. These findings suggest LLMs encode structural knowledge of physical systems—the same inductive bias LOAM leverages for effective observation and action design.
>
> **Response to W2, Q2: Sensitivity Analysis on LLM Backbones**
>
> In the revised manuscript, we additionally evaluate LOAM with GPT-5-mini, GPT-5-nano, and Qwen3-Coder-30B (open-weight). We observe performance comparable to that achieved with GPT-5, indicating that LOAM is not strongly dependent on a specific LLM. The corresponding ablation results have been added to Section 6.4 of the revised paper.
>
> **Response to W3, Q1: Other Domains**
>
> We have additionally evaluated LOAM on tasks in NVIDIA Isaac Lab, which provides a different environment from the HumanoidBench tasks used in the main experiments.
> In this environment, we assess on three representative tasks: Lift-Cube, Open-Drawer, and Repose-Cube. Across these tasks, both LOAM and LOAM-Race achieve promising performance, indicating that LOAM is not restricted to HumanoidBench and can transfer to other domains beyond our primary setting. For details, please refer to Sections 6.3 of the revised paper, where we include the corresponding setup and experimental results.
>
> On the other hand, vision-based tasks are outside the scope of our work, as feature extraction from images is already handled effectively by existing CNN/ViT-based encoders or by methods such as ExploRLLM. Nonetheless, integrating LOAM with standard visual encoders could be a promising direction for future research.
>
> **Response to W4, Q3: Related Work**
>
> We clarify that LOAM-Race is the first method to efficiently mitigate quality diversity in LLM-designed observation and action models while maintaining the total number of timesteps for a standard single-model training. The manuscript has been revised accordingly.
> Prior works (Eureka, Text2Reward, LESR) focus on reward generation, with LESR appending LLM-generated features to predefined, hand-crafted observations. In contrast, LOAM designs observation and action models from scratch—a less explored direction. We have expanded the Related Work section accordingly.
>
> We also clarify that LOAM and ExploRLLM differ substantially in both their space construction and target domains :
> - Space construction: ExploRLLM operates within a pre-defined space consisting of `[image_crops, object_positions, end_effector_state]` and uses VLMs as online perception modules to extract feature values. LOAM instead generates functions that construct observation and action spaces from scratch
> - Target domain: ExploRLLM focuses on visual pick-and-place tasks, whereas LOAM addresses broader continuous control problems (locomotion, manipulation) using non-visual inputs.
> We have added ExploRLLM to the Related Work section.
>
> **Response to W5: Real-World Applicability and Sim-to-Real Gap**
>
> Note that LOAM is a framework for general RL. Once an observation-action model (or a reward model, in related work) is produced, an RL algorithm is trained on top of it. This stage occurs prior to deployment. Accordingly, our focus is aligned with prior work on generation of environment models for improving sample efficiency, rather than with sim-to-real research. Sim-to-real studies address challenges that arise when transferring a policy trained in simulation to the real world.
>
> [1] Liang, Jacky, et al. "Code as policies: Language model programs for embodied control." arXiv preprint arXiv:2209.07753 (2022).
>
> [2] Gerstmayr, Johannes, Peter Manzl, and Michael Pieber. "Multibody models generated from natural language." Multibody System Dynamics 62.2 (2024): 249-271.

---

### Author Response · Authors · 2025-11-26
**Summary of Paper Revisions**

We have revised the manuscript throughout to improve clarity and presentation. All revisions in the updated manuscript are highlighted in blue. The major changes are summarized below:
- Refinement of Section 3 & 4.2 (Response to Reviewer A8cU): To address the concern regarding the "overly detailed" prompt descriptions in Section 3, we condensed the section by replacing Figure 2 with a summary table and moving the full prompt details to the Appendix. We utilized the saved space to significantly expand Section 4.2, providing a more detailed explanation of the LOAM-Race mechanism and its resource allocation strategy.
- Refinement of Section 5 : We have expanded the Related Work section to include a more detailed discussion of the papers suggested by the reviewers.
- New Section 6.3: Additional experiments on Isaac Lab (Response to Reviewers 17hf, snyz, 4CzN, A8cU, fque): We added a new section presenting experimental results on NVIDIA Isaac Lab.
- New Section 6.4: Ablation Studies on LLM Backbones & Acquisition Functions (Response to Reviewers snyz, 17hf): We added ablation studies analyzing the effect of different LLM backbones (including GPT-5-mini/nano and open-source Qwen3-Coder-30B) and confidence estimation methods (Bayesian Ridge vs. Gaussian Process) in LOAM-Race.

---

### Author Response · Authors · 2025-11-26
**Common Response**

For clarity, we outline our contribution more explicitly below.

### **Contribution of LOAM**

When applying RL to a new task, the first step is to formalize it as an MDP or POMDP. In practice, experts must understand the task, determine which aspects of the environment are relevant, and then manually design the observation, action, and reward. Because this effort must be repeated for every new task and demands substantial expertise, it often becomes a major bottleneck in deploying RL to new tasks.

Recent LLM-based efforts aim to reduce this manual burden, but most have focused on automating reward design or reward shaping. In contrast, the equally important problem of observation-action model design has received comparatively little attention. These two problems are complementary: reward design shapes the learning objective, whereas observation-action design determines the interface through which the agent perceives and interacts with the environment.

We propose LOAM to address this gap. The goal of LOAM is to automate the part that human designers traditionally perform: given a natural language description of the task and the available variable information, LOAM uses an LLM to generate observation and action models suitable for downstream RL training.

### **Contribution of LOAM-Race**
Because LLM-generated models can vary in quality across runs, we introduce LOAM-Race to address this variability. LOAM-Race is designed to achieve two goals simultaneously: (1) stabilize the final task performance, while (2) maintain the same total number of timesteps $T$ as a standard single-model training.

Prior works address this issue by generating multiple candidate models and selecting the best-performing one. However, identifying the optimal candidate imposes a significant computational burden, as it typically necessitates training each candidate policy to convergence to accurately assess its performance.
In contrast, LOAM-Race estimates the future potential of a candidate model without training its policy to convergence, enabling efficient model selection with significantly reduced computational cost.

To the best of our knowledge, LOAM-Race is the first method to propose an efficient learning method that mitigates quality diversity in LLM-designed observation and action models while maintaining the same total number of training timesteps $T$ for a standard single-model training.

### **Relation to Reward Design Methods**
Prior studies primarily focus on reward generation or reward modeling.
In contrast, LOAM targets a different and less explored component: designing the observation and action models. These reward-focused approaches are therefore complementary rather than overlapping with LOAM. Extending LOAM to jointly design rewards is a promising direction for future research but is beyond the scope of this work.

### **Scope of Our Work**
Note that LOAM is a framework for general RL. Once an observation-action model (or a reward model, in related work) is produced, an RL algorithm is trained on top of it. This stage occurs prior to deployment. Accordingly, our focus is aligned with prior work on generation of environment models for improving sample efficiency, rather than with sim-to-real research. Sim-to-real studies address challenges that arise when transferring a policy trained in simulation to the real world.

### **Additional Experiment1 : Effect of the LLM Backbone**
We have included additional experimental results using other LLMs, including smaller models (GPT-5-mini and GPT-5-nano) and an open-source model (Qwen3-Coder-30B), on three HumanoidBench tasks (stand, walk, run), as shown in Figure 6. Across all tasks, these LLMs achieve performance comparable to GPT-5, indicating that LOAM can operate effectively with a wide range of LLMs.

### **Additional Experiment2 : Isaac Lab Tasks**
We have additionally evaluated LOAM on tasks in NVIDIA Isaac Lab, which provides a different environment from the HumanoidBench tasks used in the main experiments.
In this environment, we assess on three representative tasks: Lift-Cube, Open-Drawer, and Repose-Cube. Across these tasks, both LOAM and LOAM-Race achieve promising performance, indicating that LOAM is not restricted to HumanoidBench and can transfer to other domains beyond our primary setting.

Detailed experimental results have been included in Section 6.3 and 6.4.

---

### Author Response · Authors · 2025-12-02
**Brief Overview to AC**

We sincerely thank the ACs and reviewers. Following the conference’s recent guidance, we provide here a brief overview of our work and the major revisions.

**1. What LOAM Does: Automating Observation and Action Model Design**

The first step in applying RL to a new task is formalizing it as an MDP or POMDP. In practice, this requires domain experts to manually identify relevant variables and design appropriate observation and action spaces—a process that is often a major bottleneck.

LOAM aims to automate this critical step. Given a natural-language task description and an extensive set of observable features in environment (i.e., raw observations $O\_{raw}$), and their descriptions, LOAM uses an LLM to generate executable Python code for the observation and action models.

**2. Positioning Relative to Related Work**

Recent research on "LLM for RL" has predominantly focused on **reward design** (e.g., Eureka[1], Text2Reward[2]), effectively automating the *learning objective*. In contrast, LOAM addresses the comparatively unexplored problem of **observation and action model design**, which determines the *interface* through which the agent perceives and interacts with the environment. We posit that these two directions are complementary. LOAM fills the gap in the automated RL pipeline by handling the interface design, and it can be seamlessly integrated with existing reward generation methods.

Some prior works have explored related ideas, such as introducing conceptual abstractions[3] or limited to partial automation[2]. **To the best of our knowledge, LOAM is the first framework to achieve fully automated, code-level generation directly from raw observation without human intervention.**  For example, in HumanoidBench, LOAM uses `mj.data` as raw observation, which contains the full MuJoCo simulation state (e.g., joint positions, velocities, body frames), and in Isaac Lab, LOAM utilizes `RigidObjectData` as raw observation.

Regarding the relationship with ExploRLLM[4], as highlighted by Reviewer 17hf, we clarify that the two approaches differ substantially in both space construction and target domains. While *ExploRLLM* utilizes VLMs as online perception modules within a pre-defined space (primarily for visual pick-and-place tasks), LOAM generates observation and action functions from scratch based on $O\_{raw}$. This capability allows LOAM to address broader continuous control problems—such as simultaneous whole-body locomotion and manipulation.

**3. Why LOAM-Race? Efficient Handling of LLM Variability**

LLM-generated outputs inherently exhibit quality diversity. While prior works address this by training multiple candidates to convergence and selecting the best one (incurring high computational costs), we propose **LOAM-Race** for achieving robust selection with the same sample efficiency as standard single-model training.

Inspired by racing algorithms[5]—which efficiently identify the optimal candidate by dynamically allocating computational resources based on statistical estimates of future potential—LOAM-Race applies an Optimism in the Face of Uncertainty strategy to evaluate candidates in parallel and adaptively allocate resources. This method identifies high-performing models **within the same total number of timesteps (T) as a standard single-model training**, significantly improving sample efficiency compared to the methods that generate and then select.

For instance, while LESR[2] consumes approximately 15.4T due to separate pilot evaluations and training, LOAM-Race completes the entire process within 1T.

**4.  Why HumanoidBench? Overcoming the Limits of Manual Design in Complex Control**

We selected HumanoidBench because it represents the frontier of continuous control complexity, requiring simultaneous whole-body control for diverse tasks(locomotion and manipulation). However, manually designing optimal interfaces for such a wide range of tasks given comprehensive set of observable features, $O_{raw}$, is prohibitively expensive. Consequently,  the benchmark defaults to a fixed, task-agnostic observation—using generic qpos and qvel of the robot and objects (if present) for all tasks—due to the prohibitive cost of manual tuning.

This setup mirrors RL deployment: practitioners often face vast sensory data ($O_{raw}$) yet require distinct feature engineering for diverse tasks. Because HumanoidBench encapsulates this exact scalability challenge, it serves as the ideal testbed to validate our approach. Our results confirm that LOAM autonomously transforms raw observation $O_{raw}$ into efficient, task-specific interfaces, proving that automated design is a necessity for solving high-dimensional control problems.

To further validate the generality of our approach, we conducted additional experiments on Isaac Lab during the rebuttal phase, confirming that LOAM's automated interface design transfers effectively across different environments.

---

> ### Author Response · Authors · 2025-12-02
> **Major Revision**
>
> While we have provided detailed point-by-point responses to each reviewer’s specific comments in the individual threads, we summarize below the major revisions and additional experiments that address the shared concerns raised across multiple reviews:
>
> **1. Demonstrated Generalization beyond HumanoidBench (Addressing Reviewer 17hf, snyz, 4CzN, A8cU)**
>
> To address concerns regarding domain overfitting, we **extended our evaluation to NVIDIA Isaac Lab**. We tested LOAM on three representative tasks (*Lift-Cube, Open-Drawer, Repose-Cube*). The results demonstrate that LOAM successfully discovers effective policies without task-specific tuning, confirming that our framework generalizes across different environments (See Section 6.3).
>
> **2. Ablation Studies on Robustness and Design Choices (Addressing Reviewer 17hf, snyz)**
>
> - LLM Robustness: We tested GPT-5-mini, GPT-5-nano, and the open-weights Qwen3-Coder-30B. All achieved performance comparable to the flagship GPT-5, confirming that LOAM-Race is cost-effective and not strictly bound to proprietary models (See Section 6.4 and Appendix F.1).
> - Acquisition Function: We compared our Bayesian Ridge Regression (BRR) acquisition function against a Gaussian Process Regression (GPR) baseline. The results confirm that BRR yields competitive performance with superior computational efficiency, justifying its selection for LOAM-Race (See Section 6.4 and Appendix F.2).
>
> **3. Clarified Efficiency and Selection Mechanism of LOAM-Race (Addressing Reviewer 17hf, snyz, A8cU)**
>
> To address the questions raised by several reviewers and ensure a precise understanding of the LOAM-Race mechanism, we have expanded Section 4.2 to provide a more detailed and intuitive explanation. Furthermore, new results in Appendix F.3 demonstrate that LOAM-Race achieves superior stability and sample efficiency compared to iterative refinement methods, effectively mitigating LLM variability without increasing the total training timesteps.
>
> **4. Expanded Related Work and Refined Manuscript Structure (Addressing Reviewer 17hf, A8cU)**
>
> We have reorganized the manuscript to improve readability and contextualize our contributions. Specifically, we **condensed the prompt descriptions in Section 3** to allow for a more in-depth discussion of the LOAM-Race algorithm (as suggested by A8cU). Additionally, we expanded **Section 5 (Related Work)** to include relevant literature suggested by the reviewers clearly delineating LOAM’s unique position in designing observation and action models from scratch compared to vision-based or reward-focused methods.
>
>
> **References**
>
> [1] Ma et al. Eureka: Human-Level Reward Design via Coding Large Language Models. In International Conference on Learning Representations. 2024.
>
> [2] Wang et al. LLM-Empowered State Representation for Reinforcement Learning. In International Conference on Machine Learning. 2024.
>
> [3] Park et al. Automatic Environment Shaping Is The Next Frontier in RL. In International Conference on Machine Learning. 2024.
>
> [4] Ma et al. ExploRLLM: Guiding Exploration in Reinforcement Learning with Large Language Models. 2025 IEEE International Conference on Robotics and Automation. 2025.
>
> [5] Oded Maron and Andrew Moore. Hoeffding Races: Accelerating Model Selection Search for Classification and Function Approximation. Advances in Neural Information Processing Systems. 1993.

---

### Meta-Review · Area_Chair_Crcj · 2026-01-07

**Summary:**

This paper introduces **LOAM** (LLM-based design of Observation and Action Models), a framework that leverages Large Language Models to automate the design of observation and action spaces in Reinforcement Learning (RL). While previous work has focused on reward design, LOAM targets the interface through which agents perceive and interact with the environment. To address the variability in LLM-generated code, the authors propose **LOAM-Race**, which evaluates multiple candidate models in parallel and adaptively allocates training resources to the most promising ones using an "Optimism in the Face of Uncertainty" strategy.

### Strengths

- **Novel Focus**: Addresses an understudied but critical bottleneck in RL—the manual engineering of observation and action representations.

- **Empirical Performance**: Demonstrates significant improvements (up to 3x faster learning) on the HumanoidBench benchmark compared to handcrafted models and existing baselines like LESR.

- **Efficiency**: LOAM-Race successfully identifies high-performing models within the same total training budget () as a standard single-model training.

- **Generalization**: Rebuttal experiments on NVIDIA Isaac Lab show that the framework can transfer across different simulation environments.

### Weaknesses

- **Limited Theoretical Grounding**: The work is largely empirical, lacking a formal analysis of why LLM-generated designs improve representation quality or exploration.

- **Dependence on Specification**: Reviewers noted that the framework’s success depends heavily on human-provided task descriptions and access to pre-existing environment variables, which may limit its utility in truly "raw" or undefined scenarios.


- **Sim-to-Real Gap**: There is no validation on physical hardware, leaving questions about how these generated models would handle real-world noise, delays, or physical constraints.

- **Prompt Engineering Concerns**: Some reviewers suggested the prompts are heavily engineered with domain-specific guidance, potentially contradicting the claim of full automation.

### Final Recommendation: Reject

While LOAM presents a compelling and timely application of LLMs to RL environment design with strong empirical results, the submission is ultimately recommended for **rejection**. The primary concerns center on the **conceptual depth** of the automation and the **practical applicability** beyond simulation. Reviewers felt that the framework acts more as a "code translator" for detailed human specifications rather than a truly autonomous design agent. Furthermore, the lack of real-world validation or a rigorous theoretical framework to explain the performance gains makes the contribution feel primarily incremental and empirical at this stage.

**Reviewer Concerns:**

## Addressed Concerns

- **Generalization:** The authors added evaluations on NVIDIA Isaac Lab tasks (Lift-Cube, Open-Drawer, Repose-Cube) to demonstrate cross-domain applicability.



- **LLM Robustness:** New results show LOAM works effectively with diverse models including GPT-5-mini, GPT-5-nano, and open-source Qwen3-Coder-30B.


- **Cost Efficiency:** Detailed analysis shows the total cost (approx. $0.74 per task) and time (20 minutes) are negligible compared to RL training.


- **LOAM-Race Clarity:** The authors clarified the racing mechanism with detailed resource allocation explanations and pseudocode.


## Outstanding Concerns

- **Depth of Automation:** Reviewers remain concerned that the framework "automates translation" of human specifications rather than performing conceptual task design.

- **Reward Dependency:** The framework still relies on access to pre-existing, human-scripted reward signals or logic, limiting its use in truly unknown scenarios.

- **Prompt Engineering:** Concerns persist that the prompts are heavily engineered with domain-specific guidance, potentially shifting manual effort from coding to prompt tuning.

- **Theoretical Grounding:** The submission lacks a formal or theoretical analysis explaining why LLM-generated designs improve representation quality.

- **Real-World Gap:** There is no physical robot validation or substantive analysis of how these models handle the noise and delays inherent in sim-to-real transfer.

**Reviewer Scores:**

### Positive Stances

- **Reviewer 17hf**: Rated **6**. They acknowledged the novel contribution of automating observation/action models and accepted the author's rebuttal regarding generalization to Isaac Lab and robustness across different LLMs.


- **Reviewer 4CzN**: Rated **6**. They were impressed by the strong empirical results on HumanoidBench and the effectiveness of the "reach" task where baselines failed.

### Negative/Skeptical Stances (Score: 4)

- **Reviewer fque**: Rated **4**. They remained skeptical, arguing that the system is essentially "translating" detailed human specifications into code rather than performing autonomous conceptual design.


- **Reviewer snyz**: Rated **4**. They raised concerns about the lack of real-robot validation and the potential costs of the racing mechanism, though authors provided a cost analysis of ~$0.74 per task in response.


- **Reviewer A8cU**: Rated **4**. They criticized the lack of iterative refinement in the code generation and felt the LOAM-Race mechanism was not clearly described in the initial draft.

---

### Decision · Program_Chairs · 2026-01-26

Reject